

# Classification of all $\mathcal{N} \geq 3$ moduli space orbifold geometries at rank 2

**Philip C. Argyres[1]⋆, Antoine Bourget[2]† and Mario Martone[3]‡**

**1** University of Cincinnati, Physics Department, Cincinnati OH 45221
**2** Theoretical Physics Group, The Blackett Laboratory,
Imperial College London, Prince Consort Road London, SW7 2AZ, UK
**3** University of Texas, Austin, Physics Department, Austin TX 78712

⋆ philip.argyres@gmail.com, † a.bourget@imperial.ac.uk, ‡ mariomartone@utexas.edu

## Abstract

We classify orbifold geometries which can be interpreted as moduli spaces of four-dimensional $\mathcal{N} \geq 3$ superconformal field theories up to rank 2 (complex dimension 6). The large majority of the geometries we find correspond to moduli spaces of known theories or discretely gauged version of them. Remarkably, we find 6 geometries which are not realized by any known theory, of which 3 have an $\mathcal{N} = 2$ Coulomb branch slice with a non-freely generated coordinate ring, suggesting the existence of new, exotic $\mathcal{N} = 3$ theories.



# 1   Introduction

Theoretical physicists' wild dream of mapping the space of quantum field theories, even when restricted to unitary, local, and Poincaré-invariant ones, is probably unattainable. But add in enough supersymmetry and the dream becomes much tamer, discrete structures emerge, and enumerating them seems within reach. Here we take a step in this direction in the case of $\mathcal{N} \geq 3$ supersymmetric field theories in 4 dimensions. In particular, we carry out a classification of the possible moduli space geometries of such theories with rank less than or equal to 2.

In many ways $\mathcal{N} = 3$ theories are an ideal fit for the classification task: they are constrained enough that it seems possible to carry out a complete classification, but unconstrained enough that the answer obtained is non-trivial. Indeed, we find here unexpected results. We carry out the analysis by analyzing the moduli space of vacua, $\mathcal{M}$,[1] of such theories. We consider the existence of a space $\mathcal{M}$ which can be consistently interpreted as the moduli space of vacua of an $\mathcal{N} = 3$ theory as strong evidence for the existence of such a theory. A similar approach turned out to be very successful in the classification of $\mathcal{N} = 2$ rank-1 geometries [1–4]. Even if not all of these geometries turn out to be associated to a field theory, they nevertheless constrain the possible set of such field theories. Conversely, we do not assume that there is necessarily a unique field theory corresponding to a given moduli space geometry. The moduli space geometry encodes only a small part of the conceivable properties of a field theory, so it is a priori unreasonable to assume that it is enough to completely determine the field theory. Indeed, even in the $\mathcal{N} = 2$ rank-1 case mentioned above, where the geometries classified contained much more information (since they represented whole families of deformations by relevant parameters), there were found to be cases [5] where more than one known field theory corresponds to the same geometry.

Even so, as with any classification claim, there is some fine print, which we can organize as three assumptions:

1.  $\mathcal{M}$ has rank $\leq 2$.

2.  $\mathcal{M}$'s associated Dirac pairing is principal.

3.  $\mathcal{M}$ is an orbifold.

---

[1]A clarification on notation, throughout the paper we will indicate as $\mathcal{M}$ the $3r$ complex dimensional moduli space of vacua of the theory, by $\mathcal{C}$ its $\mathcal{N} = 2$ $r$ complex dimensional Coulomb branch slice and by $\mathcal{H}$ its $2r$ complex dimensional Higgs branch slice, where $r$ is the rank of the theory.

Table 1: We list the orbifold geometries passing all our constraints. The Group column labels the discrete group orbifolding $\mathbb{C}^2$. The action on $\mathbb{C}^2$ is specified in Tables 2 and 3 using a Du Val label. The definition of the groups is given in the caption of those tables. If the geometry is freely generated, the column $\Delta$ CB gives the degrees (dimensions) of the generators; when it is a complete intersection, it gives the degrees of the three generators subject to one relation (gray shade). Some of the geometries are associated with known theories, which can be simple $\mathcal{N}=4$ (white), product theories (yellow; the rank 1 theories are labeled by their Kodaira class, see Table 5.) or $\mathcal{N}=3$ theories obtained from S-folds or discrete gaugings (blue; the discrete gauging of theory $\mathcal{T}$ by $\mathbb{Z}_k$ is denoted $[\mathcal{T}]_{\mathbb{Z}_k}$). Geometries which have no known realization are shaded in green. The fifth column indicates whether $\mathcal{N}=3$ enhances to $\mathcal{N}=4$; for product theories, we treat the two factors separately. Finally the last column gives the central charges. For geometries associated to multiple theories (see section 5), we report the highest value of the corresponding central charges.

| $|\Gamma|$ | Group | $\Delta$ CB | CFT realization | $\mathcal{N}=4$ | $4c=4a$ |
|---|---|---|---|---|---|
| \multicolumn{6}{c}{$\mathcal{N} \geq 3$ **Rank-2 Orbifold geometries** $\mathbb{C}^2/\mu_\tau(\Gamma)$} |
| 1 | $\mathbb{1}$ | $1, 1$ | $I_0 \times I_0$ | $\checkmark \times \checkmark$ | 2 |
| 2 | $\mathbb{Z}_2$ | $1, 2$ | $I_0 \times I_0^*$ | $\checkmark \times \checkmark$ | 4 |
| | | $2, 2, 2$ | $[U(1) \times U(1)\ \mathcal{N}=4]_{\mathbb{Z}_2}$ | $\checkmark$ | 2 |
| 3 | $\mathbb{Z}_3$ | $1, 3$ | $I_0 \times IV^*$ | $\checkmark \times \times$ | 6 |
| 4 | $\mathbb{Z}_2 \times \mathbb{Z}_2$ | $2, 2$ | $I_0^* \times I_0^*$ | $\checkmark \times \checkmark$ | 6 |
| | $\mathbb{Z}_4$ | $1, 4$ | $I_0 \times III^*$ | $\checkmark \times \times$ | 8 |
| | | $2, 4, 4$ | $[U(1) \times U(1)\ \mathcal{N}=4]_{\mathbb{Z}_4}$ | $\checkmark$ | 2 |
| 6 | $\mathbb{Z}_2 \times \mathbb{Z}_3$ | $2, 3$ | $I_0^* \times IV^*$ | $\checkmark \times \times$ | 8 |
| | Weyl($\mathfrak{su}(3)$) | $2, 3$ | $A_2\ \mathcal{N}=4$ | $\checkmark$ | 8 |
| | $\mathbb{Z}_6$ | $1, 6$ | $I_0 \times II^*$ | $\checkmark \times \times$ | 2 |
| 8 | $\mathbb{Z}_2 \times \mathbb{Z}_4$ | $2, 4$ | $I_0^* \times IV^*$ | $\checkmark \times \times$ | 10 |
| | Weyl($\mathfrak{so}(5)$) | $2, 4$ | $\mathfrak{so}(5)\ \mathcal{N}=4$ | $\checkmark$ | 10 |
| 9 | $\mathbb{Z}_3 \times \mathbb{Z}_3$ | $3, 3$ | $IV^* \times IV^*$ | $\times \times \times$ | 10 |
| 12 | $\mathbb{Z}_2 \times \mathbb{Z}_6$ | $2, 6$ | $I_0^* \times II^*$ | $\checkmark \times \times$ | 14 |
| | $\mathbb{Z}_3 \times \mathbb{Z}_4$ | $3, 4$ | $IV^* \times III^*$ | $\times \times \times$ | 12 |
| | Weyl($G_2$) | $2, 6$ | $G_2\ \mathcal{N}=4$ | $\checkmark$ | 14 |
| 16 | $\mathbb{Z}_4 \times \mathbb{Z}_4$ | $4, 4$ | $III^* \times III^*$ | $\times \times \times$ | 14 |
| | $SD_{16}$ | $4, 6, 8$ | *No known $\mathcal{T}_\mathcal{M}$ exists* | $\times$ | ? |
| | $M_4(2)$ | $4, 8, 8$ | | $\times$ | ? |
| | Weyl($\mathfrak{so}(5)$)$\rtimes \mathbb{Z}_2$ | $4, 4$ | $[\mathfrak{so}(5)\ \mathcal{N}=4]_{\mathbb{Z}_2}$ | $\times$ | 10 |
| 18 | $\mathbb{Z}_3 \times \mathbb{Z}_6$ | $3, 6$ | $IV^* \times II^*$ | $\times$ | 16 |
| | Weyl($\mathfrak{su}(3)$)$\rtimes \mathbb{Z}_3$ | $3, 6$ | $[A_2\ \mathcal{N}=4]_{\mathbb{Z}_3}/\mathcal{N}=3$ S-fold | $\times$ | 16 |
| 24 | $\mathbb{Z}_4 \times \mathbb{Z}_6$ | $4, 6$ | $III^* \times II^*$ | $\times \times \times$ | 18 |
| | $G(6,3,2)$ | $4, 6$ | *No known $\mathcal{T}_\mathcal{M}$ exists* | $\times$ | 18 |
| | Weyl($\mathfrak{so}(5)$)$\rtimes \mathbb{Z}_3$ | $6, 6, 12$ | $[\mathfrak{so}(5)\ \mathcal{N}=4]_{\mathbb{Z}_3}$ | $\times$ | 10 |
| 32 | $G(4,1,2)$ | $4, 8$ | $\mathcal{N}=3$ S-fold | $\times$ | 22 |
| 36 | $\mathbb{Z}_6 \times \mathbb{Z}_6$ | $6, 6$ | $II^* \times II^*$ | $\times \times \times$ | 22 |
| | Weyl($\mathfrak{su}(3)$)$\rtimes \mathbb{Z}_6$ | $6, 6$ | $[A_2\ \mathcal{N}=4]_{\mathbb{Z}_6}$ | $\times$ | 8 |
| | $Dic_3 \times \mathbb{Z}_3$ | $6, 12, 12$ | | $\times$ | ? |
| 48 | $ST_{12}$ | $6, 8$ | *No known $\mathcal{T}_\mathcal{M}$ exists* | $\times$ | 26 |
| 72 | $G(6,1,2)$ | $6, 12$ | | $\times$ | 34 |

Caption:

| | |
|---|---|
| Product of rank-1 theories | Known discrete gauging or S-folds |
| Discrete gauging of $U(1)^2\ \mathcal{N}=4$ | Theories with no known realization |

The first two assumptions are for technical convenience: we are confident that the approach to the classification problem described here is equally applicable to higher ranks and non-principal Dirac pairings (though it may not be technically easy to implement). The third assumption is central to our approach. In order to discuss its significance, we first describe the

main features of $\mathcal{N}=3$ moduli space geometry.

The generic low-energy physics on the moduli space $\mathcal{M}$ of a rank-$r$ theory, is that of $r$ free supersymmetric massless photons, thus its geometry is a generalized and more constrained version of the special Kähler geometry[2] enjoyed by $\mathcal{N}=2$ Coulomb branches (CBs). This $\mathcal{N}=3$ CB geometric structure is called a *triple special Kähler* (TSK) structure, and has been introduced and analyzed in [7]. More specifically a TSK geometry corresponding to a rank-$r$ $\mathcal{N}=3$ field theory is a $3r$ complex dimensional complex variety which is metrically flat almost everywhere. Away from its complex co-dimension 3 metric singularities, $\mathcal{M}$ is covered by a family of *special coordinates* which are flat complex coordinates whose monodromies are in the group of electromagnetic (EM) duality transformations, $\mathrm{Sp}_D(2r,\mathbb{Z})$, which is the group of transformations preserving the rank-$2r$ EM charge lattice, $\Lambda$, and the antisymmetric pairing, $D : \Lambda \times \Lambda \to \mathbb{Z}$, appearing in the Dirac quantization condition.[3] Also, since all $\mathcal{N}=3$ field theories with an $\mathcal{N}=3$ field theory UV fixed point are superconformal field theories [8], $\mathcal{M}$ also has a complex scaling symmetry from the action of the spontaneously broken dilatation and $\mathrm{U}(1)_R$ symmetries, as well as an $\mathrm{SU}(3)_R$ isometry.

As almost everywhere flat spaces whose flat coordinates are linearly related by finite group transformations, $\mathcal{M}$ is "almost" an orbifold. As explained via examples in [7], such an $\mathcal{M}$ can fail to be an orbifold because, even locally, identification by a group element can fail to correspond to dividing by a group action. But, the resulting non-orbifold TSK spaces have a field content when interpreted as $\mathcal{N}=3$ SCFTs which is unusual, and may be unphysical [8,9]; see [7] for a critical discussion. One characteristic of these flat non-orbifold moduli spaces is that the scaling dimensions of their chiral ring operators are not integers. Such flat non-orbifold geometries do occur (and, indeed, are common) in $\mathcal{N}=2$ CBs.

Accepting the orbifold assumption, it follows [7] that the moduli space of vacua of a rank-$r$ $\mathcal{N}=3$ field theory is a $3r$ complex dimensional variety, $\mathcal{M}$, which can be globally written as $\mathcal{M} \equiv \mathcal{M}_\Gamma \cong \mathbb{C}^{3r}/\Gamma$, with $\Gamma$ finite.[4] $\mathcal{N}=3$ supersymmetry further constrains $\Gamma$ and its action in various ways [7]. First, $\mathcal{M}_\Gamma$ has a $\mathbb{CP}^2$ of inequivalent complex structures and the orbifold action, $\rho(\Gamma)$, of $\Gamma$ on $\mathbb{C}^{3r}$ depends on the specific choice of the complex structure on $\mathcal{M}_\Gamma$. Second, the $\mathrm{SU}(3)_R$ isometry on $\mathcal{M}_\Gamma$ implies that the $\rho(\Gamma)$ action descends to an $r$-dimensional "slice", $\mathcal{C}_\Gamma = \mathbb{C}^r/\Gamma$, corresponding to an $\mathcal{N}=2$ CB subvariety of $\mathcal{M}_\Gamma$, and the analysis of $\mathcal{C}_\Gamma$ suffices to reconstruct the geometric structure of $\mathcal{M}_\Gamma$. Thus we will often state the results of a given geometry $\mathcal{M}_\Gamma$ in terms of its $r$ dimensional CB $\mathcal{C}_\Gamma$. Third, the admissible finite groups $\Gamma$ are crystallographic point groups preserving an integral symplectic form $D$ [7,10], so $\Gamma \subset \mathrm{Sp}_D(2r,\mathbb{Z})$ (more details below). Fourth, the technical assumption (2) that $D$ is principal just means that by a lattice change of basis it can be put in the standard symplectic form $D = \begin{pmatrix} 0 & \mathbb{1}_r \\ -\mathbb{1}_r & 0 \end{pmatrix}$, and so $\Gamma \subset \mathrm{Sp}(2r,\mathbb{Z})$. The list of physically consistent $\mathcal{M}_\Gamma$ can be further constrained by studying their refined Hilbert series $H_{\mathcal{M}_\Gamma}$. The first few terms of $H_{\mathcal{M}_\Gamma}$ give useful information about the operator content of the putative theory $\mathcal{T}_\mathcal{M}$ realizing a particular $\mathcal{M}_\Gamma$. In some cases, which are shaded in red in Table 2-4, we can argue that

---

[2]See, e.g., [6] for a review.

[3]At first it might appear inconsistent for a complex co-dimension 3 singular locus to give rise to monodromies. But these monodromies don't arise from path linking the singular locus but rather from the TSK patching condition [7].

[4]A word on notations is in order. The letter $\Gamma$ denotes the finite group we use to characterize orbifold geometries. When there is no risk of confusion, we write simply $\Gamma$ for the various representations of this (abstract) group. However, we sometimes use more precise notations, namely:

- $M(\Gamma)$ for the finite subgroup of $\mathrm{Sp}(2r,\mathbb{Z})$;

- $\mu_\tau(\Gamma)$ for the subgroup of $U(r)$ involved in the Coulomb branch slice orbifold, $\mathbb{C}^r/\Gamma \equiv \mathbb{C}^r/\mu_\tau(\Gamma)$;

- $\rho_\tau(\Gamma)$ for the subgroup of $U(3r)$ involved in the full orbifold, $\mathbb{C}^{3r}/\Gamma \equiv \mathbb{C}^{3r}/\rho_\tau(\Gamma)$.

All these group morphisms are defined in section 2.

such operator content is unphysical. We then discard the corresponding $\mathcal{M}_\Gamma$ despite it being a consistent TSK.

For rank $r = 1$ the result of classifying such orbifolds is fairly easy as the only finite subgroups of $\mathrm{SL}(2, \mathbb{Z}) \cong \mathrm{Sp}(2, \mathbb{Z})$ are $\mathbb{Z}_2$, $\mathbb{Z}_3$, $\mathbb{Z}_4$ and $\mathbb{Z}_6$. The corresponding orbifold geometries correspond to the known rank-1 $\mathcal{N} \geq 3$ geometries [11, 12], with the $\mathbb{C}^3/\mathbb{Z}_2$ corresponding to the moduli space geometry of the $\mathrm{SU}(2)$ $\mathcal{N} = 4$ theory. Here we extend the analysis to rank 2. The results are summarized in Table 1 as well as more systematically listed in Tables 2, 3 and 4, where for each orbifold we report both the $2 \times 2$ matrix of low energy EM couplings, $\tau^{ij}$, characteristic of the TSK metric geometry, as well as detailed data about the complex geometry of $\mathcal{C}_\Gamma \subset \mathcal{M}_\Gamma$. For $r = 2$, the picture that arises is far richer than the $r = 1$ case: many geometries that we find do not correspond to known physical theories and some of these new geometries exhibit interesting and novel properties.

One of the main results of our analysis is that the coordinate ring of many of the admissible geometries is not a freely generated ring even when restricted to the $r$-complex dimensional slice $\mathcal{C}_\Gamma$. It follows in these cases that as a complex space $\mathcal{C}_\Gamma$ is an algebraic variety not isomorphic to $\mathbb{C}^r$. This fact has a straightforward physical interpretation in a field theory corresponding to $\mathcal{M}_\Gamma$. The coordinate ring of $\mathcal{C}_\Gamma$ is the CB chiral ring of the $\mathcal{N} = 3$ theory relative to a choice of an $\mathcal{N} = 2$ subalgebra of the $\mathcal{N} = 3$ algebra. It is by now well known that CB chiral rings can be non-freely generated [13–16], though the only known examples thus far arise after gauging a discrete symmetry group. We conjecture that some of the geometries which we find correspond to theories with a non-freely generated CB chiral ring that does not arise from discrete gauging. It, of course, remains to be proven that such geometries do in fact correspond to physical theories. A variety of possible extra checks which can be performed are listed at the end of the paper.

It is worth remarking that we don't perform our classification by directly studying the finite subgroups of $\mathrm{Sp}(4, \mathbb{Z})$ which would give rise to consistent rank-2 TSK geometries, but instead by using a related property of $\mathcal{M}_\Gamma$ that follows from $\mathcal{N} = 3$ supersymmetry. This property is that, for any rank $r$, the matrix $\tau^{ij}$ of EM couplings on $\mathcal{M}_\Gamma$, which by standard arguments is an element of the fundamental domain of the Siegel upper half space, $\mathfrak{H}_r$, is fixed by the action of $\Gamma \subset \mathrm{Sp}(2r, \mathbb{Z})$. So another way of proceeding is to first classify all possible fixed points of elements of $\mathrm{Sp}(2r, \mathbb{Z})$ in $\mathfrak{H}_r$ and the subgroups which fix them. As stated, this may not seem to simplify the classification problem. But to our surprise E. Gottschling [17, 18] classified all fixed points in $\mathfrak{H}_2$. With a bit of extra work, both because the papers are in German and because Gottschling only classifies the maximal $\Gamma \subset \mathrm{Sp}(2r, \mathbb{Z})$ fixing a given $\tau^{ij}$, we are able to use the results in [17, 18] to fully characterize all rank-2 $\mathcal{N} = 3$ orbifold geometries.

The paper is organized as follows. In the next section we quickly review the definition and the main properties of TSK geometries. We will make the conscious choice of sacrificing pedagogy for conciseness, and generously refer to [7] for the details. We do, however, carry out the explicit construction of the TSK geometry of the moduli space of vacua of $\mathrm{SU}(3)$ $\mathcal{N} = 4$ theory as an illustrative example. Section 3 describes in some detail how the classification is performed and systematically discusses the list of geometries which we find. In section 4 we analyze in detail the refined Hilbert series of the geometries we constructed. After a review of $\mathcal{N} = 3$ superconformal representation theory, we identify how to put the two together to set more stringent physical constraints on the geometries we find. A discussion of the physics of the allowed geometries, along with a specification of which geometries are new and which correspond to known theories is given in section 5. This section also contains a discussion about the allowed possibilities of discrete gauging in the case of product theories. This discussion is, to our knowledge, new and perhaps of interest to the reader. We conclude and present a number of interesting possible follow up directions. A series of appendices collect some technical material about Du Val groups and Hilbert series.

## 2 Orbifold geometries and $\mathcal{N} = 3$ preserving conditions

As mentioned above, a systematic analysis of the moduli space geometry of $\mathcal{N} = 3$ SCFTs is given in [7], including a discussion of possible non-orbifold geometries which we are not going to consider here. In this section we will just summarize the most important points to remind the reader how to construct an orbifold TSK structure on $\mathcal{M}_\Gamma$ and justify our classification strategy.

First, a note on terminology. $\Gamma$ will refer to the orbifold group as an abstract finite group, and we will use homomorphisms $\rho_\tau : \Gamma \to \mathrm{GL}(3r, \mathbb{C})$, $\mu_\tau : \Gamma \to \mathrm{GL}(r, \mathbb{C})$, and $M : \Gamma \to \mathrm{Sp}(2r, \mathbb{Z})$ to denote its action on various spaces defined below.

Our classification strategy rests on the following assertions. If a moduli space of an $\mathcal{N} = 3$ SCFT with a principal Dirac pairing is an orbifold then

1. $\mathcal{M}_\Gamma = \mathbb{C}^{3r}/\rho_\tau(\Gamma)$ where the $\mathbb{C}^{3r}$ are the special coordinates $a_i^I$, $i = 1, \ldots, r$ and $I = 1, 2, 3$, and $\rho_\tau : \Gamma \to \mathrm{GL}(3r, \mathbb{C})$ is a homomorphism of a finite group $\Gamma$ into $\mathrm{GL}(3r, \mathbb{C})$;

2. $\Gamma$ acts on $\mathbb{C}^{3r} = \mathbb{C}^3 \otimes \mathbb{C}^r$ as $\rho_\tau(\Gamma) \subset \mathbb{1}_3 \otimes \mathrm{GL}(r, \mathbb{C})$, so that $\rho_\tau = \mu_\tau \oplus \mu_\tau \oplus \mu_\tau$ for some homomorphism $\mu_\tau : \Gamma \to \mathrm{GL}(r, \mathbb{C})$;

3. there is a homomorphism $M : \Gamma \to \mathrm{Sp}(2r, \mathbb{Z})$ since the monodromies also act by multiplication by $\mathrm{Sp}(2r, \mathbb{Z})$ matrices on the $2r$-component vector $(a_D^{Ii}, a_i^I)$ (for all $I$) where $a_D^{Ii} := \tau^{ij} a_j^I$ and $\tau^{ij}$ is a point in the Siegel upper half-space $\mathfrak{H}_r$.

We will briefly review the justification of these assertions in the next subsection.

Write $M \in \mathrm{Sp}(2r, \mathbb{Z})$ in $r \times r$ blocks as $M = \begin{pmatrix} A & B \\ C & D \end{pmatrix}$. Then the $\Gamma$ action in assertion (3) gives in an obvious matrix notation

$$M(\Gamma) \ni M : \begin{pmatrix} a_D \\ a \end{pmatrix} \mapsto \begin{pmatrix} a_D' \\ a' \end{pmatrix} = \begin{pmatrix} A a_D + B a \\ C a_D + D a \end{pmatrix}. \tag{1}$$

But since $a_D = \tau a$ and $a_D' = \tau a'$ this means

$$\begin{pmatrix} \tau \\ 1 \end{pmatrix} a' = \begin{pmatrix} A\tau + B \\ C\tau + D \end{pmatrix} a, \tag{2}$$

which is only consistent if

$$\tau = (A\tau + B)(C\tau + D)^{-1}, \tag{3}$$

i.e., if $\tau$ is fixed by the usual fractional linear action of $M(\Gamma) \subset \mathrm{Sp}(2r, \mathbb{Z})$ on $\mathfrak{H}_r$. Furthermore, (1) induces the $\mathrm{GL}(r, \mathbb{C})$ action

$$\mu_\tau(\Gamma) \ni \mu_\tau(M) : a \mapsto (C\tau + D)a, \tag{4}$$

which, via assertion (2), gives the $\mathrm{GL}(3r, \mathbb{C})$ orbifold action of assertion (1). Notice that (4) is only a group homomorphism if $\tau \in \mathrm{Fix}(\Gamma)$, that is if (3) is satisfied for all elements in $M(\Gamma)$.

Thus we see that an orbifold is an $\mathcal{N} = 3$ moduli space for a principally polarized SCFT if and only if $\Gamma$ is isomorphic to a finite subgroup $M(\Gamma) \subset \mathrm{Sp}(2r, \mathbb{Z})$ which fixes a $\tau \in \mathfrak{H}_r$. Furthermore, the orbifold then has the form

$$\mathcal{M}_\Gamma = \mathbb{C}^{3r}/\rho_\tau(\Gamma) = \mathbb{C}^{3r}/(\mathbb{1}_3 \otimes \mu_\tau(\Gamma)), \tag{5}$$

with $\mu_\tau$ completely determined by $M(\Gamma)$ and the fixed $\tau$. In the complex structure of $\mathbb{C}^{3r}$ in which the $a_j^I$ are holomorphic coordinates, the $\rho_\tau(\Gamma)$ action in (5) acts holomorphically as

does the $U(3)_R$ isometry. We will see, however, that this complex structure is *not* the complex structure of $\mathcal{M}_\Gamma$ which is determined by the supersymmetry. A given $M(\Gamma) \subset Sp(2r, \mathbb{Z})$ may have more than one fixed point $\tau$, and the choice of $\tau$ is part of the moduli space geometry. This is the reason for the $\tau$ subscript on the homomorphism $\mu_\tau$.

We can thus classify all possible $\mathcal{N} = 3$ orbifold moduli spaces by classifying finite subgroups of $Sp(2r, \mathbb{Z})$ (up to conjugation) with fixed points in $\mathfrak{H}_r$. Note that all finite subgroups of $Sp(2r, \mathbb{Z})$ fix at least one point in $\mathfrak{H}_r$; see e.g., section 3.2 of [10]. Thus one way of proceeding is to find all finite subgroups of $Sp(2r, \mathbb{Z})$ and then compute their fixed points. But we will instead do things in the reverse order by finding all possible fixed points of the $Sp(2r, \mathbb{Z})$ action on $\mathfrak{H}_r$ and then characterize the finite subgroups which fix them. We will show how to do this when $r \leq 2$ in section 3, and now turn to justifying assertions (1)–(3).

## 2.1 Metric geometry of $\mathcal{M}_\Gamma$

At a generic point on the moduli space of vacua $\mathcal{M}_\Gamma$ of an $\mathcal{N} = 3$ SCFT, the theory is described by $r$ free massless vector multiplets in the IR. These $\mathcal{N} = 3$ vector multiplets have $U(1)^r$ gauge fields — making $\mathcal{M}_\Gamma$ a Coulomb branch — as well as $3r$ complex scalar fields $a_i^I$, $I = 1, 2, 3$ and $i = 1, .., r$, which transform in the $r$-fold direct sum of $\mathbf{3}_1$ representations of the $U(3)_R$ symmetry group. From the point of view of an $\mathcal{N} = 2$ subalgebra, this $\mathcal{N} = 3$ vector multiplet is a free $\mathcal{N} = 2$ vector multiplet plus a free massless neutral hypermultiplet. $\mathcal{N} = 2$ supersymmetry implies that these massless bosonic fields have an IR effective Lagrangian

$$\mathcal{L}_{\text{bosonic}} = \text{Im}\big[\tau^{ij}(a)\big(\partial a_i^I \cdot \partial \overline{a}_{Ij} + \mathcal{F}_i \cdot \mathcal{F}_j\big)\big], \tag{6}$$

where $\mathcal{F}_i$ are the self-dual $U(1)$ gauge field strengths and $\tau^{ij}$ takes values in the Siegel upper-half space $\mathfrak{H}_r$ — i.e., $\tau^{ij} = \tau^{ji}$ and $\text{Im}\tau^{ij} > 0$. Furthermore, an $\mathcal{N} = 2$ selection rule [19] forbids the vector multiplet metric and the hypermultiplet metric from depending on the same fields, so

$$\tau^{ij} = \text{ constant}. \tag{7}$$

The scalar kinetic term in (6) induces the metric

$$g = (\text{Im}\tau^{ij}) da_i^I d\overline{a}_{Ij} \tag{8}$$

on $\mathcal{M}_\Gamma$. Since $\tau^{ij}$ is constant, the metric is flat, and the $a_i^I$ are flat coordinates. The vevs of the $a_i^I$ are called *special coordinates* on $\mathcal{M}_\Gamma$. On overlaps of special coordinate patches on $\mathcal{M}_\Gamma$, since they are flat the special coordinates are related by linear transition functions plus possible constant shifts. Note that this description holds at generic points on $\mathcal{M}_\Gamma$, but there may be curvature singularities along a subspace $\mathcal{V} \subset \mathcal{M}_\Gamma$; we will see below that this subspace is of at least complex co-dimension 3.

Non-zero vevs of the $a_i^I$ spontaneously break the conformal invariance and the $U(3)_R$ symmetry, so $\mathcal{M}_\Gamma$ will have a scaling symmetry and a $U(3)_R$ isometry. The scale invariance and overall $U(1)_R$ factor combine to make $\mathcal{M}_\Gamma$ a complex cone. The tip of the cone is the origin of any and all special coordinate patches, and corresponds to the unique conformal vacuum.

If $\mathcal{M}_\Gamma$ is an orbifold then assertions (1) and (2) now follow. Since the special coordinate patches all have their origins in common, transition functions must be linear transformations of the $3r$ complex special coordinates. This gives assertion (1) with $\rho_\tau(\Gamma) \subset GL(3r, \mathbb{C})$. The existence of a $U(3)_R$ isometry then implies that the orbifold identifications must commute with the $U(3)_R$ action on the special coordinates, giving assertion (2).

The (massive) states at a generic point on $\mathcal{M}_\Gamma$ are labeled by their vector $\mathbf{p} \in \mathbb{Z}^{2r}$ of magnetic and electric charges under the low energy $U(1)^r$ gauge group. These vectors span a rank-$2r$ charge lattice. The Dirac quantization condition defines a non-degenerate, integral,

and skew bilinear pairing $\langle \mathbf{p}, \mathbf{q} \rangle := \mathbf{p}^T D \mathbf{q} \in \mathbb{Z}$. By a change of charge lattice basis, the integral skew-symmetric matrix $D$ can be brought to the unique canonical form $D = \varepsilon \otimes \Delta$ where $\varepsilon$ is the $2 \times 2$ unit antisymmetric tensor and $\Delta := \text{diag}\{\delta_1, \ldots, \delta_r\}$ is characterized by $r$ positive integers $\delta_i$ satisfying $\delta_i | \delta_{i+1}$. The electric-magnetic (EM) duality group, $\text{Sp}_D(2r, \mathbb{Z}) \subset \text{GL}(2r, \mathbb{Z})$, is the subgroup of the group of charge lattice basis changes which preserves the Dirac pairing. If all the $\delta_i = 1$ we call the Dirac pairing *principal* and $\text{Sp}_D(2r, \mathbb{Z}) = \text{Sp}(2r, \mathbb{Z})$. Theories with non-principal Dirac pairings are allowed, a priori, but it may be that they are only relative field theories [20]. From now on we will assume a principal polarization simply because it is technically easier to work with $\text{Sp}(2r, \mathbb{Z})$ than with $\text{Sp}_D(2r, \mathbb{Z})$. It is an interesting question to extend our classification to non-principal polarizations.

Just as in $\mathcal{N} = 2$ theories [21], EM duality transformations act linearly on $2r$-component complex vectors $\boldsymbol{\sigma} = (a_D, a)^T$ made up of the special coordinates, $a_i^I$, and the dual special coordinates

$$a_D^{Ii} := \tau^{ij} a_j^I. \tag{9}$$

We call $\boldsymbol{\sigma}$ the special Kähler section on an $\mathcal{N} = 2$ CB [22]. In the case of $\mathcal{N} = 3$ theories there are now three such $2r$-component special Kähler sections,

$$\boldsymbol{\sigma}^I := \begin{pmatrix} a_D^{Ii} \\ a_i^I \end{pmatrix}, \qquad I = 1, 2, 3, \tag{10}$$

giving $\mathcal{M}_\Gamma$ a *triple special Kähler* (TSK) structure.

In particular, EM duality transformations, $M \in \text{Sp}(2r, \mathbb{Z})$, act on the TSK sections by matrix multiplication, $\boldsymbol{\sigma}^I \mapsto M \boldsymbol{\sigma}^I$, and upon traversing a closed path, $\gamma$, in $\mathcal{M}_\Gamma$, the sections may transform by an EM duality transformation, $M_\gamma \in \text{Sp}(2r, \mathbb{Z})$. The set of all such EM monodromies generates a finite subgroup $M(\Gamma) \subset \text{Sp}(2r, \mathbb{Z})$ [7]. Since the orbifold identifications give rise to monodromies of the special coordinates, this gives us assertion (3).

The locus of metric singularities $\mathcal{V} \subset \mathcal{M}_\Gamma$ occur where charged states become massless. This can only happen when the BPS lower bound on their mass vanishes. In an $\mathcal{N} = 3$ theory the BPS bound on the mass of a state with EM charges $\mathbf{p}$ is $m \geq Z^I(\mathbf{p}) \overline{Z}_I(\mathbf{p})$ where $Z^I$ is the $SU(3)_R$ triplet of complex central charges of the $\mathcal{N} = 3$ algebra. In the low energy theory on $\mathcal{M}_\Gamma$ we have $Z^I(\mathbf{p}) := \mathbf{p}^T \boldsymbol{\sigma}^I$. Metric singularities $\mathcal{V}$ can thus only occur where the central charges vanish for some $\mathbf{p}$: $Z^I(\mathbf{p}) = 0$ for $i = 1, 2, 3$. Thus $\mathcal{V}$ is of complex co-dimension 3 in $\mathcal{M}_\Gamma$. In the orbifold geometries studied here this follows automatically: metric singularities in the orbifold occur at fixed points of the $\text{GL}(3r, \mathbb{C})$ orbifold group action on $\mathbb{C}^{3r}$, but by assertion (2) this action lies only in a $\text{GL}(r, \mathbb{C}) \subset \text{GL}(3r, \mathbb{C})$ so its fixed point locus is of co-dimension 3.

## 2.2 Complex geometry of $\mathcal{M}_\Gamma$

So far we have described the metric geometry of $\mathcal{M}_\Gamma$, but not specified its complex structure. The complex structure is important for identifying the chiral ring of the underlying SCFT, and for understanding how $\mathcal{N} = 2$ Coulomb and Higgs branches are embedded in the $\mathcal{N} = 3$ moduli space.

The complex structure of $\mathcal{M}_\Gamma$ is determined by picking one left-handed supercharge in the $\mathcal{N} = 3$ algebra and calling the complex scalars which are taken to left-handed Weyl spinors by the action of that supercharge the holomorphic coordinates on $\mathcal{M}_\Gamma$. The $\mathcal{N} = 3$ supersymmetry variations of the vector multiplet fields then imply [7] that the special coordinates are *not* holomorphic coordinates on $\mathcal{M}_\Gamma$. Rather, out of each $SU(3)_R$ triplet, two can be taken to be holomorphic and the third anti-holomorphic. Thus, for example,

$$(z_i^1, z_i^2, z_{3i}) := (a_i^1, a_i^2, \overline{a}_{3i}), \qquad i = 1, \ldots, r, \tag{11}$$

can be taken as holomorphic coordinates on $\mathcal{M}_\Gamma$.[5] Note that (11) implies that the $U(1)_R$ isometry acts holomorphically on $\mathcal{M}_\Gamma$, but that the $SU(3)_R$ isometry does not.

Choosing an $\mathcal{N} = 2$ subalgebra of the $\mathcal{N} = 3$ algebra corresponds to choosing a minimally embedded $SU(2)_R \subset SU(3)_R$. Then the subspace, $\mathcal{C}_\Gamma \subset \mathcal{M}_\Gamma$, fixed by this $SU(2)_R$ isometry is the $\mathcal{N} = 2$ Coulomb branch. For example, an $\mathcal{N} = 2$ subalgebra compatible with the complex structure (11) on $\mathcal{M}_\Gamma$ is one in which $(a_i^2, a_i^3)$ transform as a doublet of the $SU(2)_R$ and $a_i^1$ as a singlet. Then the associated CB "slice" of $\mathcal{M}_\Gamma$ is $\mathcal{C}_\Gamma = \{a_i^2 = a_i^3 = 0\}$. Since its complex coordinate are the special coordinates $a_i^1$, it inherits an $\mathcal{N} = 2$ special Kähler structure from the $\mathcal{N} = 3$ TSK structure. Because the $SU(3)_R$ is a global isometry of $\mathcal{M}_\Gamma$, this description of $\mathcal{C}_\Gamma$ valid in a special coordinate patch extends to all of $\mathcal{C}_\Gamma$.

In the case where $\mathcal{M}_\Gamma$ is an orbifold, identifying the complex structure globally is straight forward. For instance, in the complex structure (11) take $(z_i^1, z_i^2, z_{3i}) \in \mathbb{C}^{3r}$ so that given the homomorphism $\mu_\tau : \Gamma \to GL(r, \mathbb{C})$ defined in (4), then $\mathcal{M}_\Gamma$ is, as a complex space, the orbifold

$$\mathcal{M}_\Gamma \equiv \mathbb{C}^{3r} / \mu_\tau(\Gamma) \oplus \mu_\tau(\Gamma) \oplus \overline{\mu_\tau}(\Gamma). \tag{12}$$

This should be contrasted with its description as the orbifold (5), where its $U(3)_R$ isometry is manifest but its complex structure is not.

Note that in the case where $\rho_\tau(\Gamma) = \overline{\rho_\tau}(\Gamma)$ is real, the two descriptions coincide. This is precisely the case where the isometry group is enhanced to $SO(6)_R$ [7], and corresponds to $\mathcal{M}_\Gamma$ satisfying the conditions of $\mathcal{N} = 4$ supersymmetry.

An $\mathcal{N} = 2$ CB slice of $\mathcal{M}_\Gamma$ is then clearly the orbifold

$$\mathcal{C}_\Gamma \equiv \mathbb{C}^r / \mu_\tau(\Gamma), \tag{13}$$

with the metric

$$g = (\operatorname{Im}\tau^{ij}) da_i^1 d\overline{a}_{1j} \tag{14}$$

inherited from (8). In the orbifold case we can also embed an $\mathcal{N} = 2$ Higgs branch $\mathcal{H}_\Gamma \subset \mathcal{M}_\Gamma$ by going to the $2r$-dimensional $a_i^1 = 0$ slice. This gives

$$\mathcal{H}_\Gamma \equiv \mathbb{C}^{2r} / \mu_\tau(\Gamma) \oplus \overline{\mu_\tau}(\Gamma). \tag{15}$$

This is a hyperKähler cone with an $SU(2)_R$ non-holomorphic isometry, and a $U(1)_F$ tri-holomorphic "flavor" isometry [7].

Since the $\mu_\tau(\Gamma)$ action on $\mathbb{C}^r$ given by (4) is enough to reconstruct the entire TSK structure on $\mathcal{M}_\Gamma$, we will often discuss the $\mathcal{N} = 2$ CB orbifold $\mathcal{C}_\Gamma$ rather than $\mathcal{M}_\Gamma$ when it will allow for a more direct and less cumbersome discussion. Such $\mathcal{N} = 2$ CB orbifolds in the case where $\mu_\tau(\Gamma)$ is a complex reflection group were considered in [10].

Generally $\mu_\tau(\Gamma)$ does not act freely on $\mathbb{C}^r$. The locus $\mathcal{V} \subset \mathbb{C}^r$ of points fixed by at least one non-identity element $\mu_\tau(\Gamma)$ is the locus of metric non-analyticities. Since $\mu_\tau(\Gamma)$ acts holomorphically, $\mathcal{V}$ is a complex subvariety of $\mathcal{C}_\Gamma$, generically of complex co-dimension 1. Furthermore, unless $\mu_\tau(\Gamma)$ is a complex reflection group, $\mathcal{C}_\Gamma$ is not isomorphic to $\mathbb{C}^r$ as a complex variety, as its coordinate ring is not freely generated [23, 24]. In such cases, a subvariety $\mathcal{V}_{cplx} \subset \mathcal{V}$ will also have complex singularities [14, 25]. We emphasize that the generic point in $\mathcal{V}$ has a metric non-analyticity (curvature singularity) but a smooth complex structure.

---

[5]This expression is valid, in particular, for the complex structure induced by the $Q_\alpha^3$ supercharge. There is a $\mathbb{CP}^2$-worth of inequivalent ways of embedding one left-handed supercharge in the $\mathcal{N} = 3$ algebra, so there is, in fact, a $\mathbb{CP}^2$ of inequivalent complex structures on $\mathcal{M}_\Gamma$ [7].

## 2.3   SU(3) $\mathcal{N} = 4$ **superYang-Mills: an explicit example**

To get a better sense of the rather abstract discussion in the previous section, let's now carry out the construction outlined above in a simple example, namely the SU(3) $\mathcal{N} = 4$ sYM theory. We will particularly focus on computing the fixed locus of the group action, $\mathcal{V}$, compute from there the corresponding BPS states which become massless along $\mathcal{V}$ and compare to the expected result from the weak coupling limit of the sYM theory finding perfect agreement. A physical interpretation of the other moduli space geometries we construct in section 3 will be given in section 5.

The moduli space of vacua, $\mathcal{M}_\Gamma(\mathfrak{g})$, of an $\mathcal{N} = 4$ sYM theory with gauge Lie algebra $\mathfrak{g}$, is parameterized by the vevs of the complex Cartan subalgebra scalar fields, $a_i^I$ for $I = 1, 2, 3$ and $i = 1, \ldots, r = \text{rank}(\mathfrak{g})$. The geometry gets no quantum corrections but is orbifolded by any gauge identifications of a given Cartan subalgebra of the gauge Lie algebra. These identifications are given by the finite Weyl group, $\mathcal{W}(\mathfrak{g})$, of the Lie algebra, thus in this case $\Gamma = \mathcal{W}(\mathfrak{g})$. The Weyl group acts as a real crystallographic reflection group on the real Cartan subalgebra, i.e., via orthogonal transformations, $w \in \text{O}(r, \mathbb{R}) \subset \text{GL}(r, \mathbb{C})$, with respect to the Killing metric on the Cartan subalgebra. From our discussion above it then follows that in the special case of $\mathcal{N} = 4$ theories, the orbifold action (12) can be written as $\mathbb{1}_3 \otimes w$, where $\mathbb{1}_3$ denotes the $3 \times 3$ identity matrix.

The $\mathcal{N} = 4$ sYM theory can be viewed as an $\mathcal{N} = 2$ theory with respect to a choice of an $\mathcal{N} = 2$ subalgebra of the $\mathcal{N} = 4$ superconformal algebra. From this point of view, the $\mathcal{N} = 4$ moduli space decomposes into an $\mathcal{N} = 2$ Coulomb branch $\mathcal{C}_\Gamma(\mathfrak{g})$ (an $r$ complex-dimensional special Kähler space) and an $\mathcal{N} = 2$ Higgs branch $\mathcal{H}_\Gamma(\mathfrak{g})$ (an $r$ quaternionic-dimensional hyperKähler space) which are each subspaces of a $3r$ complex dimensional enhanced Coulomb branch [4]. The geometries of these Coulomb and Higgs branches are induced from the geometry of $\mathcal{M}_\Gamma(\mathfrak{g})$ in the obvious way, replacing the $\mathbb{1}_3 \otimes w$ with $w$ and $\mathbb{1}_2 \otimes w$ respectively. Thus

$$\mathcal{C}_\Gamma(\mathfrak{g}) = \mathbb{C}^r / \mathcal{W}(\mathfrak{g}) \qquad \mathcal{W}(\mathfrak{g}) \subset \text{O}(r, \mathbb{R}) \subset \text{GL}(r, \mathbb{C}). \tag{16}$$

We take the holomorphic coordinates on $\mathbb{C}^r$ to be $z_i = a_i^1$, $i = 1, \ldots, r$.

In this case the complex structure of $\mathcal{C}_\Gamma(\mathfrak{g})$ turns out to be very simple: as a complex space the $\mathcal{N} = 2$ Coulomb branch is isomorphic to $\mathbb{C}^r$ and thus has no complex singularities. This result follows from the powerful Chevalley-Shepard-Todd (CST) theorem [23, 24] and the fact that $\mathcal{W}(\mathfrak{g})$ is a (real) complex reflection group. $\mathcal{C}_\Gamma(\mathfrak{g})$ of course still has metric singularities (non-analyticities) at the orbifold fixed-point loci which we will discuss shortly. Furthermore, since the action of $\mathcal{W}(\mathfrak{g})$ on $\mathcal{C}_\Gamma(\mathfrak{g})$ is via orthogonal transformations, $\mathcal{W}(\mathfrak{g})$ preserves a real symmetric bilinear form $s$. This implies that there is always a degree 2 polynomial $P_2 = z_i s^{ij} z_j$ which is invariant under the action of the orbifold group and can be thus chosen as one of the global holomorphic coordinates on $\mathcal{C}_\Gamma(\mathfrak{g})$. This is the distinguishing feature of geometries associated to $\mathcal{N} = 4$ theories. In fact, the existence of a dimension 2 generator of the coordinate ring of $\mathcal{C}_\Gamma(\mathfrak{g})$ implies the existence of a dimension 2 generator of the CB chiral ring. It is a standard result of superconformal representation theory that such a dimension 2 CB multiplet contains an exactly marginal operator which is identified with the gauge coupling for $\mathfrak{g}$.

GL(2, $\mathbb{C}$) **group action and singular locus.**   Let us now specify the previous general discussion to the case of a SU(3) $\mathcal{N} = 4$ sYM theory. In this case $\mathcal{W}(\mathfrak{su}(3)) \cong S_3$ and its action on $\mathbb{C}^2$ giving rise to $\mathcal{C}_\Gamma(\mathfrak{su}(3)) = \mathbb{C}^2 / \mathcal{W}(\mathfrak{su}(3))$ is generated by

$$w_1 := \begin{pmatrix} -1 & -1 \\ 0 & 1 \end{pmatrix}, \qquad w_2 := \begin{pmatrix} 0 & 1 \\ 1 & 0 \end{pmatrix}, \tag{17}$$

which act linearly on $(z_1, z_2) \in \mathbb{C}^2$.

As mentioned above, $\mathcal{C}_\Gamma(\mathfrak{su}(3))$ is isomorphic to $\mathbb{C}^2$ as an algebraic variety. In other words we expect that its coordinate ring is a polynomial ring in two variables, and, furthermore, to have one degree (dimension) two generator. Using standard Hilbert series techniques (reviewed below) and the explicit group action in (17), we can do the computation explicitly and obtain that coordinate ring of $\mathcal{C}_\Gamma(\mathfrak{su}(3))$ is in fact a polynomial ring generated by

$$u = \tfrac{1}{6}\left[ z_1^2 + z_2^2 + (z_1 + z_2)^2 \right], \qquad v = \tfrac{1}{2}\left[ z_1 z_2(-z_1 - z_2) \right]. \tag{18}$$

(The normalization is arbitrary and is chosen only to simplify a later formula.) This shows that $\mathcal{C}_\Gamma(\mathfrak{su}(3))$ as a complex variety has no singularities.

Let us now compute the metric singularities by studying the fixed loci of the group (17). Calling $\mathcal{V}_{1,2}$, the fixed loci of $w_{1,2}$, a straightforward calculation shows that:

$$\mathcal{V}_1: \quad z_1 - z_2 = 0, \qquad \mathcal{V}_2: \quad 2z_1 + z_2 = 0. \tag{19}$$

Since $\mathcal{V}_1$ and $\mathcal{V}_2$ are connected by a $\mathcal{W}(\mathfrak{su}(3))$ transformation, the singular locus has only one connected component in this case which we will simply call $\mathcal{V}$. This is even more obvious writing $\mathcal{V}$ in terms of the globally defined coordinates on $\mathcal{C}_\Gamma(\mathfrak{su}(3))$. Then in the $(u, v)$ coordinates we can write $\mathcal{V} \equiv \mathcal{V}_1 \equiv \mathcal{V}_2$

$$\mathcal{V}: \quad u^3 = v^2. \tag{20}$$

This result coincides with the known single trefoil knot singularity of the SU(3) $\mathcal{N} = 4$ theory [26].

**GL(4, $\mathbb{Z}$) action and symplectic form.** Before discussing the monodromy transformation picked up by the special coordinates on $\mathcal{C}_\Gamma(\mathfrak{su}(3))$ encircling $\mathcal{V}$, let's take a short detour describing how to construct an Sp(2r, $\mathbb{Z}$) representation for rank-$r$ crystallographic complex reflection groups.

Rank-$r$ crystallographic complex reflection groups act irreducibly on $\mathbb{C}^r$ and are thus naturally defined in GL$(r, \mathbb{C})$. But it can be shown that, by choosing an appropriate basis in $\mathbb{C}^r$, they can actually be defined on GL$(r, \mathbb{Z}[\sqrt{-d}]) \subset$ GL$(r, \mathbb{C})$, where $d$ is a square free integer and $\mathbb{Z}[\sqrt{-d}]$ is a degree two extension of $\mathbb{Z}$ [10, 27]. By representing $\sqrt{-d}$ by $\begin{pmatrix} 0 & 1 \\ -d & 0 \end{pmatrix}$ we can construct a natural representation in GL$(2r, \mathbb{Z})$. Furthermore by averaging over the group we can construct an invariant Hermitian form with coefficients in $\mathbb{Z}[\sqrt{-d}]$ whose imaginary part provides an integral skew-symmetric form $D$ which is by construction preserved by the group action [7, 10]. In general $D$ is not principal, but when it is then there is a natural representation of the crystallographic complex reflection group in Sp$(2r, \mathbb{Z}) \subset$ GL$(2r, \mathbb{Z})$.

The situation is considerably simpler for a Weyl group $\mathcal{W}(\mathfrak{g})$. $\mathcal{W}(\mathfrak{g})$ is a real crystallographic reflection group and acts on the root lattice, $\Lambda^{\mathfrak{g}}_{\text{root}}$, which is a lattice of rank $r$, not $2r$. In other words, in the appropriate basis, the $\mathcal{W}(\mathfrak{g})$ action on $\mathbb{C}^r$ can be written as matrices $w \in$ GL$(r, \mathbb{Z})$. A consequence of that is that Weyl groups act on a one (complex) parameter family of rank $2r$ lattices, obtained by "complexifying" the root lattice

$$\Lambda^\tau_{\mathcal{W}(\mathfrak{g})} = \tau(\lambda)\Lambda^{\mathfrak{g}}_{\text{root}} \oplus \Lambda^{\mathfrak{g}}_{\text{root}}, \tag{21}$$

where $\tau(\lambda)$ is an element in the Siegel upper-half space $\mathfrak{H}_r$ acting on the base vectors of $\Lambda^{\mathfrak{g}}_{\text{root}}$ which depends on a single complex number $\lambda$ parametrizing the family of lattices. The existence of a single free parameter is a reflection of the exactly marginal operator of the corresponding $\mathcal{N} = 4$ theory. Choosing a basis in $\mathbb{C}^r$ "aligned" with the lattice (21) provides a GL$(2r, \mathbb{Z})$ representation of $\mathcal{W}(\mathfrak{g})$ which can furthermore be lifted to matrices $M_w \in$ Sp$(2r, \mathbb{Z})$, $\forall w \in \mathcal{W}(\mathfrak{g})$:

$$M_w = \begin{pmatrix} w^{-T} & 0 \\ 0 & w \end{pmatrix} \tag{22}$$

where $w^{-T}$ represents the transpose of $w^{-1}$. It is straightforward to see that (22) preserves the symplectic form

$$J = \begin{pmatrix} 0 & -\mathbb{1}_r \\ \mathbb{1}_r & 0 \end{pmatrix}, \tag{23}$$

thus $M_w \in \mathrm{Sp}(2r, \mathbb{Z})$. The action (22) is induced by the choice of the lattice in (21). In particular (22) requires that $w$ acts as $w^{-T}$ on the basis identified by the base vectors of $\tau \Lambda_{\mathrm{root}}^{\mathfrak{g}}$. In other words it is obtained by choosing a $\tau$ such that $\tau^{-1} w \tau = w^{-T}$.

It is straightforward to apply this general construction to $S_3$. We have already written the generators (17) as matrices in $\mathrm{GL}(2, \mathbb{Z})$. From (22) we can construct the generators of the $\mathrm{Sp}(4, \mathbb{Z})$ representation of $S_3$

$$M_{w_1} = \begin{pmatrix} -1 & 0 & 0 & 0 \\ -1 & 1 & 0 & 0 \\ 0 & 0 & -1 & -1 \\ 0 & 0 & 0 & 1 \end{pmatrix}, \qquad M_{w_2} = \begin{pmatrix} 0 & 1 & 0 & 0 \\ 1 & 0 & 0 & 0 \\ 0 & 0 & 0 & 1 \\ 0 & 0 & 1 & 0 \end{pmatrix}, \tag{24}$$

as well as the $\tau$ which implements the correct choice of the rank-$2r$ lattice,

$$\tau(\lambda)_{S_3} = \begin{pmatrix} \lambda & -\lambda/2 \\ -\lambda/2 & \lambda \end{pmatrix}. \tag{25}$$

Having written down the explicit expression of both the $\mathrm{GL}(2, \mathbb{C})$ and $\mathrm{Sp}(2r, \mathbb{Z})$ matrices we can explicitly check (4) in this case: $\mu_\tau(M_{w_{1,2}}) = w_{1,2}$.

**Monodromy and BPS spectrum.** We now have all the ingredients to study the special geometry of $\mathcal{C}_\Gamma(\mathfrak{su}(3))$. Using (9), (10), and (25) we can construct the special Kähler section:

$$\boldsymbol{\sigma}_{\mathfrak{su}(3)} = \begin{pmatrix} \lambda z_1 + \frac{\lambda}{2} z_2 \\ \frac{\lambda}{2} z_1 + \lambda z_2 \\ z_1 \\ z_2 \end{pmatrix}. \tag{26}$$

Notice that for convenience we have used a slightly different $\tau$ matrix to define $\boldsymbol{\sigma}_{\mathfrak{su}(3)}$ which satisfies $\tau'^{-1} w^{-T} \tau' = w$ instead. Since $\tau'$ is related to (25) by an $\mathrm{Sp}(2r, \mathbb{Z})$ transformation they define the same lattice $\Lambda_{S_3}^\tau$. We will use $\tau$ to denote the two matrices interchangeably.

Consider now a closed loop $\gamma \in \mathcal{C}_\Gamma(\mathfrak{su}(3))$ encircling $\mathcal{V}$. This is not a closed loop in $\mathbb{C}^2$, rather the end point of the loop $\mathbf{z}_1$ is related by an $S_3$ transformation to the starting point $\mathbf{z}_0$: $\mathbf{z}_1 = w_{1,2} \mathbf{z}_0$. Since we have the special Kähler section in terms of the affine coordinates (26), we immediately compute the resulting monodromy to be

$$\boldsymbol{\sigma}_{\mathfrak{su}(3)}(\mathbf{z}_0) \xrightarrow{\gamma} \boldsymbol{\sigma}_{\mathfrak{su}(3)}(w_{1,2} \mathbf{z}_0) = M_{w_{1,2}} \boldsymbol{\sigma}_{\mathfrak{su}(3)}(\mathbf{z}_0), \tag{27}$$

where we have used the fact that $\tau(\lambda)$ is fixed by the $S_3$ action: $\tau(\lambda) w = w^{-T} \tau(\lambda)$. This is a check that the Weyl group orbifold identifications induce the associated EM duality monodromies (24).

Physically we expect charged BPS states to become massless along $\mathcal{V}$ and we can use the explicit monodromy to get some insights into the low-energy physics along $\mathcal{V}$. Here we will follow an argument outlined in section 4.2 of [26]. The basic idea is that the states becoming massless at $\mathcal{V}$ are all charged under only a single low energy U(1) gauge factor: an appropriate EM duality transformation will set, say, the last two components of these charge vectors to zero. We will call the two factors $\mathrm{U}(1)_\perp$ and $\mathrm{U}(1)_\parallel$ respectively. Going to the $\mathrm{U}(1)_\perp \times \mathrm{U}(1)_\parallel$ basis factorizes the physics into a free $U(1)$ factor, $U(1)_\parallel$, with only massive states, and a non-trivial,

either conformal or IR free, rank-1 theory, the $U(1)_\perp$ factor. Understanding the spectrum of BPS states becoming massless on $\mathcal{V}$ is tantamount to understanding the non-trivial rank-1 piece.

The factorization of the physics should also be reflected in the monodromy matrix which after the EM duality transformation $U(1)^2 \to \mathrm{U}(1)_\perp \times \mathrm{U}(1)_\parallel$, acquires a very special form. In particular shuffling around the components of $\boldsymbol{\sigma}_{\mathfrak{su}(3)}$ we can choose a different, more appropriate for the purpose at hand, symplectic basis where the symplectic form is

$$J' = \begin{pmatrix} \epsilon & 0 \\ 0 & \epsilon \end{pmatrix}, \qquad \epsilon := \begin{pmatrix} 0 & 1 \\ -1 & 0 \end{pmatrix}. \tag{28}$$

Then the monodromy matrix around $\mathcal{V}$ takes the general form [26]:

$$M_{\mathcal{V}} = \begin{pmatrix} M_\perp & D \\ f(D) & \mathbb{1}_2 \end{pmatrix}. \tag{29}$$

Here $D$ and $f(D)$ are uninteresting matrices with zero determinant and the $M_\perp \in \mathrm{SL}(2,\mathbb{Z})$ is the monodromy associated with the rank-1 theory on $\mathcal{V}$ which is what we are after.

Choosing an appropriate EM duality transformation, both matrices in (24) can be written as:

$$M_{\mathcal{V}} = \left( \begin{array}{cc:cc} -1 & 0 & 0 & 1 \\ 0 & -1 & 0 & 0 \\ \hdashline 0 & -1 & 1 & 0 \\ 0 & 0 & 0 & 1 \end{array} \right), \tag{30}$$

which tells us immediately that $M_\perp = -\mathbb{1}_2$. It is a well known fact that the rank-1 theory associated with this $\mathrm{SL}(2,\mathbb{Z})$ element is the $\mathcal{N} = 4$ SU(2) theory.[6] We conclude, purely from our monodromy analysis that the theory on $\mathcal{V}$ has to be the rank-1 $\mathcal{N} = 4$ theory and the metric singularity arises where there is an enhancement of the unbroken gauge group from $\mathrm{U}(1) \to \mathrm{SU}(2)$.

To complete our analysis of the SU(3) theory we can check explicitly that the states becoming massless on $\mathcal{V}$ are in fact the gauge bosons associated to the unbroken SU(2) directions. Using the BPS bound from the central charge $Z(\mathbf{q}) = \mathbf{q}^T \boldsymbol{\sigma}_{\mathfrak{su}(3)}$ and the explicit expression, (26), for $\boldsymbol{\sigma}_{\mathfrak{su}(3)}$, we can solve for the charges of the states becoming massless on $\mathcal{V}$. From (19) we obtain that

$$\mathbf{q}_1 = (1, -1, 0, 0), \qquad \mathbf{q}_2 = (2, 1, 0, 0), \tag{31}$$

which, observing that $(q_1, q_2, 0, 0)$ and $(q_1, -q_2, 0, 0)$ are $\mathrm{Sp}(4, \mathbb{Z})$ equivalent, perfectly match the charges of the $W^\pm$ bosons under the Cartan directions in $\mathfrak{su}(3)$. Fixing a choice of simple roots, the two choices of $\mathbf{q}$'s, and thus the difference between $\mathcal{V}_1$ and $\mathcal{V}_2$, is due to whether the enhanced $\mathfrak{su}(2)$ is along one of the two simple roots ($\mathbf{q}_2$) or the third positive root of $\mathfrak{su}(3)$.

# 3 Classification of the geometries

In this section we explain how the classification of allowed $\Gamma$ is carried out and characterize the corresponding CB slice geometries $\mathcal{C}_\Gamma = \mathbb{C}^r / \mu_\tau(\Gamma)$ for $r \leq 2$. The strategy that we follow

---

[6]This statement is a bit too quick. In fact the only thing we can infer from the monodromy study is the scale CB invariant geometry associated to a given theory which does not specify the theory uniquely. It is well known that many inequivalent theories can share the same scale invariant CB geometry. In particular in this case, the $\mathcal{N} = 2$ SU(2) theory with $N_f = 4$ also gives rise to the same $-\mathbb{1}$ monodromy. In this case the existence of the $\mathcal{N} = 4$ supersymmetry implies that it cannot be the $N_f = 4$ theory. Alternatively, a study of the $\mathcal{N} = 2$-preserving mass deformations of the Coulomb branch, or of the Higgs branch sticking out of the singular locus $\mathcal{V}$ would also be able to distinguish these two possibilities.

is to first list the $\tau \subset \mathfrak{H}_r$, with $r \leq 2$, fixed by at least one $\mathrm{Sp}(2r, \mathbb{Z})$ matrix. We then determine $M(\Gamma) \subset \mathrm{Sp}(2r, \mathbb{Z})$ such that each $\tau \in \mathrm{Fix}(\Gamma)$. Then the action of $\Gamma$ on $\mathbb{C}^r$ can be easily determined by the $\mu_\tau$ map defined in (4).

Since the discussion in this section is quite technical, the reader only interested in the physics can mostly focus on subsection 3.3 where our results are summarized.

## 3.1 Mathematical preliminaries

First of all, we recall briefly a few necessary definitions and notations concerning the modular group of rank $r$ and the natural space on which it acts, the Siegel upper-half space. The rank, $r$, is a positive integer, which we will ultimately set equal to 2. But it is useful to keep it arbitrary for a while, as setting $r = 1$ in formulas helps to make the connection with the more familiar context of the standard $\mathrm{Sp}(2, \mathbb{Z}) = \mathrm{SL}(2, \mathbb{Z})$ modular group.

Let $\mathfrak{H}_r$ be the $\frac{1}{2}r(r+1)$-complex-dimensional Siegel upper half space, i.e. the space of complex symmetric $r \times r$ matrices $\tau$ with $\mathrm{Im}\,\tau$ positive definite. The rank $r$ modular group is the group $\mathrm{Sp}(2r, \mathbb{Z})$ of $2r \times 2r$ integer matrices of the form

$$M = \begin{pmatrix} A & B \\ C & D \end{pmatrix}, \tag{32}$$

which preserve the symplectic form

$$J = \begin{pmatrix} 0 & \mathbb{1}_r \\ -\mathbb{1}_r & 0 \end{pmatrix}, \tag{33}$$

meaning that $MJM^T = J$. Using the parameterization in (32), this is equivalent to the two conditions

$$AB^T \text{ and } CD^T \text{ symmetric}, \qquad \text{and} \qquad AD^T - BC^T = \mathbb{1}_r. \tag{34}$$

The modular group acts naturally on the Siegel half-space via fractional linear transformations,

$$M : \tau \mapsto (A\tau + B)(C\tau + D)^{-1}. \tag{35}$$

For $M(\Gamma) \subset \mathrm{Sp}(2r, \mathbb{Z})$ a subgroup which fixes a given $\tau \in \mathfrak{H}_r$, recall the definition (4) of the homomorphism $\mu_\tau$,

$$\mu_\tau : M(\Gamma) \to \mathrm{GL}(r, \mathbb{C}) \tag{36}$$
$$M \mapsto C\tau + D,$$

where $C, D$ are related to $M$ as in (32). Note that for (35) to be well-defined, the matrix $\mu_\tau(M)$ is non-singular. The center $\{\pm \mathbb{1}_{2r}\} \subset \mathrm{Sp}(2r, \mathbb{Z})$ acts trivially on $\mathfrak{H}_r$, so we define

$$\Delta_r = \mathrm{Sp}(2r, \mathbb{Z})/\{\pm \mathbb{1}_{2r}\}. \tag{37}$$

## 3.2 The classification strategy

In order to complete our task, the first step is to find all the possible points $\tau \in \mathfrak{H}_r$ that can be fixed points of at least one element of the modular group. The fact that not all points of $\mathfrak{H}_r$ satisfy this condition can be intuited from the rank 1 example. In that case, the fixed points are the cusps (which do not belong to $\mathfrak{H}_1$ itself, but only to an appropriate closure), and the elliptic points $\tau = i$ and $\tau = e^{2i\pi/3}$ and their (infinitely many) images under modular transformations. We avoid all these images by restricting our attention to a fundamental domain of the action of $\Delta_1$, defined for instance by the interior of the region $|\mathrm{Re}(\tau)| \leq \frac{1}{2}$ and $|\tau| \geq 1$ together with

half its boundary. For simplicity we will keep all boundaries, and call the resulting set the *fundamental region*; of course the drawback is that some points on the boundaries might have more than one image under $\Delta_1$ in the domain. This is the case for $\tau = e^{2i\pi/3}$, whose image $\tau = e^{2i\pi/3} + 1$ also lies in the fundamental region.

The general description of a fundamental region $\mathcal{F}_r \subset \mathfrak{H}_r$ of the $\Delta_r$ action on $\mathfrak{H}_r$ is a difficult problem. We refer to chapter I of [28] for a detailed presentation. Suffice it to say that the region is described by a finite number of algebraic conditions on $\tau \in \mathfrak{H}_r$. In the case which will be of interest to us, $r = 2$, the conditions for being in $\mathcal{F}_2$ can be reduced to 28 conditions. Writing

$$\tau = \begin{pmatrix} z_1 & z_3 \\ z_3 & z_2 \end{pmatrix}, \tag{38}$$

with $z_k = x_k + i y_k$, the conditions are

- $-\frac{1}{2} \leq x_k \leq \frac{1}{2}$ for $k = 1, 2, 3$ ,

- $0 \leq 2y_3 \leq y_1 \leq y_2$ ,

- $|z_k| \geq 1$ for $k = 1, 2$ ,

- $|z_1 + z_2 - 2z_3 \pm 1| \geq 1$ ,

- $|\det(Z + S)| \geq 1$ for $\pm S = \begin{pmatrix} 0 & 0 \\ 0 & 0 \end{pmatrix}, \begin{pmatrix} 1 & 0 \\ 0 & 0 \end{pmatrix}, \begin{pmatrix} 0 & 0 \\ 0 & 1 \end{pmatrix}, \begin{pmatrix} 1 & 0 \\ 0 & 1 \end{pmatrix}, \begin{pmatrix} 1 & 0 \\ 0 & -1 \end{pmatrix}, \begin{pmatrix} 0 & 1 \\ 1 & 0 \end{pmatrix}, \begin{pmatrix} 1 & 1 \\ 1 & 0 \end{pmatrix}, \begin{pmatrix} 0 & 1 \\ 1 & 1 \end{pmatrix}$ .

Now that we have defined our choice of fundamental region (at least for $r \leq 2$), we turn to finding the possible fixed points $\tau \in \mathfrak{H}_r$ under the action of subgroups of the modular group. The simplest such fixed points correspond to those left invariant by a single matrix $M \in \mathrm{Sp}(2r, \mathbb{Z})$. We will denote by $\mathfrak{H}_r(M)$ the subsets $\mathfrak{H}_r(M) \subset \mathfrak{H}_r$ of points $\tau \in \mathfrak{H}_r$ fixed by $M$. In general, $\mathfrak{H}_r(M)$ will be a complex submanifold of $\mathfrak{H}_r$, and a priori its complex dimension can be anything between 0 and $\frac{1}{2}r(r+1)$. Then given a set of matrices $\Omega \subset \mathrm{Sp}(2r, \mathbb{Z})$ — note that $\Omega$ is not not necessarily closed under matrix multiplication — we call similarly $\mathfrak{H}_r(\Omega)$ the intersection of all the $\mathfrak{H}_r(M)$ for $M \in \Omega$. Of course, $\mathfrak{H}_r(M)$ can have many disconnected components in general, since it is invariant under the action of the discrete group $\Delta_r$. It is enough to determine the connected components that intersect the fundamental region $\mathcal{F}_r$ defined above. With these notations established, our first problem can be summarized as

> *Step 1: Find all fixed point sets $\mathfrak{H}_r(\Omega)$ for $\Omega \subset \mathrm{Sp}(2r, \mathbb{Z})$*
> *which have non-empty intersection with $\mathcal{F}_r$.*

Let us illustrate this in the case of rank $r = 1$. There the possible points in $\mathcal{F}_1$ fixed by non-identity elements in $\Delta_1$ are $\tau_1 = i$, $\tau_2 = e^{2i\pi/3}$ and $\tau_3 = e^{i\pi/3}$. The elements of $\mathrm{Sp}(2, \mathbb{Z})$ which fix $\tau_i$ are the sets $\Omega_i$ given by $\Omega_1 = \{\pm \begin{pmatrix} 0 & -1 \\ 1 & 0 \end{pmatrix}\}$, $\Omega_2 = \{\pm \begin{pmatrix} 0 & 1 \\ -1 & 1 \end{pmatrix}, \pm \begin{pmatrix} 1 & -1 \\ 1 & 0 \end{pmatrix}\}$, and $\Omega_3 = \{\pm \begin{pmatrix} 0 & -1 \\ 1 & 1 \end{pmatrix}, \pm \begin{pmatrix} 1 & 1 \\ -1 & 0 \end{pmatrix}\}$. (Here, for brevity, we have left off $\pm \mathbb{1}_1$ in all these sets which trivially fixes all $\tau \in \mathfrak{H}_1$.) Then for each subset $\Omega \subset \Omega_i$, $\mathfrak{H}_1(\Omega)$ is the corresponding $\tau_i$.

For rank $r = 2$ this problem was completely solved in 1961 by E. Gottschling [17], and results in a finite list of manifolds. These complex manifolds can be parameterized by complex numbers that we denote generically by $z_1$, $z_2$ and $z_3$. For instance, to the set

$$\Omega = \left\{ \begin{pmatrix} -1 & 1 & 0 & 0 \\ -1 & 0 & 0 & 0 \\ 0 & 0 & 0 & 1 \\ 0 & 0 & -1 & -1 \end{pmatrix} \right\} \tag{39}$$

we associate the manifold of matrices of the form (38) satisfying $z_1 = z_2 = 2z_3 = \lambda$, which we denote

$$\begin{pmatrix} \lambda & \frac{1}{2}\lambda \\ \frac{1}{2}\lambda & \lambda \end{pmatrix}. \tag{40}$$

Note that the intersection of this manifold with the fundamental region described above is of lower dimension, since the conditions impose that the real part of $\lambda$ be equal to $-\frac{1}{2}$. We will not try to specify further the intersections with the fundamental region, and in the following we describe the manifolds using their parameterization, as in (40). These appear in the rightmost column in the tables below.

Now that the possible fixed points are known, the second step is to determine the subgroups of $\mathrm{Sp}(2r, \mathbb{Z})$ that leave them invariant. For any subset $S \subset \mathfrak{H}_r$, we call $\Delta_r(S)$ the maximal subgroup of $\Delta_r$ that leaves all the elements of $S$ invariant. The problem is to

*Step 2: Find all maximal subgroups $\Delta_r(S) \subset \Delta_r$ for $S \subset \mathfrak{H}_r$.*

Clearly, finding all the possible $\Delta_r(S)$ reduces to computing $\Delta_r(\mathfrak{H}_r(\Omega))$, where the $\mathfrak{H}_r(\Omega)$ have been classified above.

For rank $r = 1$ the result is easily seen to be $\Delta_1(\mathfrak{H}_1) = \{I\}$, $\Delta_1(\{\tau = i\}) = \{I, S\} \simeq \mathbb{Z}_2$, $\Delta_1(\{\tau = e^{2\pi i/3}\}) = \{I, ST, STST\} \simeq \mathbb{Z}_3$, and $\Delta_1(\{\tau = e^{\pi i/3}\}) = \{I, TS, TSTS\} \simeq \mathbb{Z}_3$. Here we have described the $\Delta_1 = \mathrm{PSL}(2, \mathbb{Z})$ elements as words in the generators $S = \begin{pmatrix} 0 & 1 \\ -1 & 0 \end{pmatrix}/\{\pm\mathbb{1}\}$, and $T = \begin{pmatrix} 1 & 1 \\ 0 & 1 \end{pmatrix}/\{\pm\mathbb{1}\}$. This task was also performed for $r = 2$ by Gottschling in a second paper the same year [18], yielding a finite list of subgroups of $\Delta_2$.

From this, it is straightforward to lift these subgroups of $\Delta_r$ to the corresponding subgroups of $\mathrm{Sp}(2r, \mathbb{Z})$ by simply "undoing" the identification in (37), thus forming a list of subgroups that we call $\mathcal{L}_r(S)$.

By definition, $\Delta_r(\mathfrak{H}_r(\Omega))$ is the largest subgroup of $\Delta_r$ that fixes $\mathfrak{H}_r(\Omega)$, but for our purposes, we also want to consider subgroups of $\Delta_r(\mathfrak{H}_r(\Omega))$ which will ultimately give the list of allowed groups we are after. Since all the groups are finite, it is in principle straightforward to enlarge $\mathcal{L}_r$ so that if $\Gamma \in \mathcal{L}_r$ then any subgroup of $\Gamma$ is also in $\mathcal{L}_r$. Thus,

*Step 3: Find all subgroups $\mathcal{L}_r(S) \subset \mathrm{Sp}(2r, \mathbb{Z})$ for $S \subset \mathfrak{H}_r$.*

Computationally, this could be a difficult task. However, the orders of the groups that come out of Gottschling's classification at rank 2 are small enough (the largest group contains 72 elements), making it possible to use a brute force enumeration algorithm.

For the rank $r = 1$ case the result is again easily obtained, though lengthy since we have to list all subgroups: $\mathcal{L}_1(\mathfrak{H}_1) = \{\Gamma_1, \Gamma_2\}$, $\mathcal{L}_1(\{\tau = i\}) = \{\Gamma_3\}$, $\mathcal{L}_1(\{\tau = e^{2\pi i/3}\}) = \{\Gamma_4, \Gamma_5, \zeta\}$, and $\mathcal{L}_1(\{\tau = e^{\pi i/3}\}) = \{\Gamma_7, \Gamma_8, \Gamma_9\}$, where $\Gamma_1 = \{I\}$, $\Gamma_2 = \{\pm I\}$, $\Gamma_3 = \{\pm I, \pm S\}$, $\Gamma_4 = \{I, -ST, STST\}$, $\Gamma_5 = \{I, ST, -STST\}$, $\Gamma_6 = \{\pm I, \pm ST, \pm STST\}$, $\Gamma_7 = \{I, -TS, TSTS\}$, $\Gamma_8 = \{I, TS, -TSTS\}$, $\Gamma_9 = \{\pm I, \pm TS, \pm TSTS\}$. Here we now use $I = \begin{pmatrix} 1 & 0 \\ 0 & 1 \end{pmatrix}$, $S = \begin{pmatrix} 0 & 1 \\ -1 & 0 \end{pmatrix}$, and $T = \begin{pmatrix} 1 & 1 \\ 0 & 1 \end{pmatrix}$ to denote elements of $\mathrm{SL}(2, \mathbb{Z})$, not $\mathrm{PSL}(2, \mathbb{Z})$. Note that $\Gamma_1$ and $\Gamma_2$ are also subgroups of the $\mathrm{SL}(2, \mathbb{Z})$ lifts of $\Delta_1(\{\tau = i\})$, $\Delta_1(\{\tau = e^{2\pi i/3}\})$, and $\Delta_1(\{\tau = e^{\pi i/3}\})$, but they are not included in the corresponding $\mathcal{L}_1$ since they already appeared as subgroups fixing the larger fixed point set $\mathfrak{H}_1$. Still, it is clear that this list of groups is highly redundant; we will eliminate this redundancy in the next step.

But first we illustrate this third step in the rank 2 case of the example (40). Gottschling computed that the subgroup of $\Delta_2$ that fixes (40) is an order 6 group, which upon lifting to

Sp$(4, \mathbb{Z})$ becomes an order 12 group generated by the two matrices[7]

$$\begin{pmatrix} -1 & 1 & 0 & 0 \\ 0 & 1 & 0 & 0 \\ 0 & 0 & -1 & 0 \\ 0 & 0 & 1 & 1 \end{pmatrix}, \begin{pmatrix} 0 & 1 & 0 & 0 \\ 1 & 0 & 0 & 0 \\ 0 & 0 & 0 & 1 \\ 0 & 0 & 1 & 0 \end{pmatrix}. \tag{41}$$

This group turns out to be isomorphic to the Weyl group of the exceptional Lie algebra $G_2$, and the corresponding geometry appears as item number 12 in Table 2. This group has 16 subgroups: the trivial group, seven isomorphic to $\mathbb{Z}_2$, one isomorphic to $\mathbb{Z}_3$, three isomorphic to $\mathbb{Z}_2^2$, one isomorphic to $\mathbb{Z}_6$, two isomorphic to $S_3$ (which is the Weyl group of $A_2$), and the full group. For each of these groups, we recompute the manifold of fixed points. For the $\mathbb{Z}_3$, the $\mathbb{Z}_6$ and the two $S_3$, we find again (40), and these appear as items number 27, 30 and 6 in the tables of the next section. For the other subgroups, we find a manifold of complex dimension $\geq 2$ which contains (40) as a submanifold, and we discard them since they will appear as subgroups of the group associated to these manifolds.

Once the list $\mathcal{L}_r$ and the associated fixed points are known, the next step is:

*Step 4: Compute the inequivalent actions on $\mathbb{C}^r$ of all $\Gamma \in \mathcal{L}_r$.*

It is a purely mechanical task to compute the action of a given $M(\Gamma) \subset \text{Sp}(2r, \mathbb{Z})$ on $\mathbb{C}^r$ via the $\mu_\tau$ homomorphism defined in (36) for any $\tau$ where $\tau$ is any element of the fixed manifold. If $\tau$ belongs to a non-zero complex dimensional locus, we do not have to compute it for each $\tau$ in the fixed manifold since as abstract groups all $\mu_\tau(\Gamma) \simeq \mu_{\tau'}(\Gamma)$. Furthermore, by definition, $G = \mu_\tau(\Gamma) \subset \text{GL}(r, \mathbb{C})$ preserves the flat metric (14) on $\mathbb{C}^r$. By a change of coordinates on $\mathbb{C}^r$ we can realize $\mu_\tau(\Gamma)$ as a finite subgroup of $U(r)$, which we will denote by $\mu(\Gamma)$ since it is independent of the choice of $\tau$ in the fixed manifold of $\Gamma$. Note that $\mu(\Gamma)$ defines the CB slice orbifold $\mathcal{C}_\Gamma = \mathbb{C}^r / \mu(\Gamma)$ as a metric and complex geometry, but does not determine the special Kähler structure of $\mathcal{C}_\Gamma$ unless the particular value of $\tau$ in the fixed manifold of $\Gamma$ is also given.

In the rank 1 case, it is immediate to see that $\Gamma_1 \simeq \mathbb{Z}_1$, $\Gamma_2 \simeq \mathbb{Z}_2$, $\Gamma_3 \simeq \mathbb{Z}_4$, $\Gamma_4 \simeq \Gamma_5 \simeq \Gamma_7 \simeq \Gamma_8 \simeq \mathbb{Z}_3$, and $\Gamma_6 \simeq \Gamma_9 \simeq \mathbb{Z}_6$ as subgroups of $U(1)$. Thus, metrically, there are just five rank-1 orbifold CB geometries, namely $\mathbb{C}/\mathbb{Z}_1 = \mathbb{C} := I_0$, $\mathbb{C}/\mathbb{Z}_2 := I_0^*$, $\mathbb{C}/\mathbb{Z}_3 := IV^*$, $\mathbb{C}/\mathbb{Z}_4 := III^*$, $\mathbb{C}/\mathbb{Z}_6 := II^*$, where we have given the names of the Kodaira types of the singularities of their associated Seiberg-Witten curves. The $I_0$ and $I_0^*$ orbifolds fix any $\tau \in \mathfrak{H}_1$, the $IV^*$ and $II^*$ orbifolds fix $\tau = e^{2\pi i/3}$, and the $III^*$ orbifold fixes $\tau = i$. This completely specifies their special Kähler geometries.

In the rank 2 case, as constructed here, the list $\mathcal{L}_2$ gives a few hundreds of distinct subgroups of $U(2)$, but not all of them give rise to physically distinct geometries since some are conjugates within $U(2)$. In order to eliminate unnecessary redundant groups, we use the fact that the $U(2)$ subgroups have been classified by Du Val [29]. We review this classification in appendix A.

### 3.3 Results

For each Du Val class of $U(2)$ subgroups, we have worked out an explicit presentation, all of this being summarized in the column "Explicit form" of Table 12. Given this, we compute three group theoretic invariants,

- the list of the orders of the elements in the group,

---

[7]The matrices which appeared in the original paper are different representatives of the same conjugacy Sp$(4, \mathbb{Z})$ class. Our choice is motivated to be consistent with the discussion in other sections of the paper.

- the sizes of the conjugacy classes (this is an integer partition of the cardinality of the group), and

- the unrefined Hilbert series of the ring of invariants given by the Molien formula.

The specification "unrefined" is to warn the reader that in this paper we will consider two different Hilbert series. The second one, which we will call instead *refined* Hilbert series, will be defined in the next section and will track more information thanks to extra fugacities given by the U(3) non-holomorphic isometry of the TSK space. We recall that the Molien formula for a finite group $G$ of matrices takes the simple form

$$H_G(t) = \frac{1}{|G|} \sum_{g \in G} \frac{1}{\det(1 - tg)} \,. \tag{42}$$

The resulting object is the graded dimension of the ring of invariants of the group $G$. In many cases, it is instructive to compute the plethystic logarithm (PLog) of this Hilbert series, defined by [30, 31]

$$\text{PLog}_G(t) = \sum_{k=1}^{\infty} \frac{\mu(k)}{k} \log\left(H_G(t^k)\right), \tag{43}$$

where $\mu$ is the Möbius function

$$\mu(k) = \begin{cases} 0 & k \text{ has one or more repeated prime factors} \\ 1 & k = 1 \\ (-1)^n & k \text{ is a product of } n \text{ distinct primes} \end{cases} \tag{44}$$

Let's briefly discuss the physics interpretation of the PLog, see [14, 32, 33] for a more in-depth discussion. The PLog of a Hilbert series "is a generating series for the relations and syzygies of the variety" in the words of [32, 33]. In simpler terms, it associates to a Hilbert series of a coordinate ring of a variety a power series in which at a given order $k$ generators of the coordinate ring appear with positive signs while relations among those generators appear with negative signs. If the variety is a complete intersection, the PLog terminates, and for non-complete intersections it is instead generally an infinite series where higher degrees count syzygies, that is relations among relations. We will see examples of these various cases shortly. It is worth reminding the reader that there exist many examples in which the Hilbert series only capture a limited amount of information about the variety and its PLog fails at correctly identifying generators, relations and syzygies of its coordinate ring.

It turns out that these three pieces of data uniquely characterize each Du Val geometry. As a consequence, for each entry in the list $\mathcal{L}_2$, we can compute the same group theoretic invariants, and read out the corresponding Du Val geometry; if two entries give the same Du Val geometry, we list the latter only once. For instance, let's compute the three invariants in the case of the order 12 group considered above:

- the orders of the elements are $\{1, 2, 2, 2, 2, 2, 2, 2, 3, 3, 6, 6\}$,

- the sizes of the conjugacy classes are $\{1, 1, 2, 2, 3, 3\}$,

- the Hilbert series of the ring of invariants is $(1 - t^2)^{-1}(1 - t^6)^{-1}$, and its PLog is $t^2 + t^6$.

This uniquely identifies the Du Val geometry $\text{DV}_3(1, 3)$.

The results for rank 2 of the computations outlined in section 3.2 are presented in the form of three tables:

Table 2: List of geometries whose holomorphic coordinate rings are freely generated. The geometry depends not just on the abstract group $\Gamma$ but on its action $\mu(\Gamma) \in \mathrm{GL}(2,\mathbb{C})$ which is determined by its Du Val class. With an abuse of notation, the direct product in the table above signifies that each factor of the product groups acts irreducibly and $\mu(\Gamma_1 \times \Gamma_2) = \mu(\Gamma_1) \oplus \mu(\Gamma_2)$. $\mathrm{ST}_n$ denotes the $n$-th Shephard-Todd group [23]. The meaning of the colors is the same as in Table 1.

| # | $|\Gamma|$ | $\Gamma$ | Du Val class | $PLog_\Gamma(t)$ | $\mathcal{O}(t^2)$ $PLog_{\mathcal{M}_\Gamma}$ | $\mathrm{Fix}(\Gamma)$ |
|---|---|---|---|---|---|---|
| 1 | 1 | 1 | $DV_1(1,1,2,1)$ | $2t$ | $0$ | $\begin{pmatrix} z_1 & z_3 \\ z_3 & z_2 \end{pmatrix}$ |
| 2 | 2 | $\mathbb{Z}_2$ | $DV_1(1,1,4,1)$ | $t+t^2$ | $\mathcal{O}_1+\mathcal{O}_2+\mathcal{O}_3$ | $\begin{pmatrix} z_1 & \frac{z_1}{2} \\ \frac{z_1}{2} & z_2 \end{pmatrix}$ |
| 3 | 3 | $\mathbb{Z}_3$ | $DV_1(1,1,6,1)$ | $t+t^3$ | $\mathcal{O}_3$ | $\begin{pmatrix} z_1 & 0 \\ 0 & e^{2\pi i/3} \end{pmatrix}$ |
| 4 | 4 | $W(\mathfrak{so}(4))=\mathbb{Z}_2\times\mathbb{Z}_2$ | $DV_1(2,2,2,1)$ | $2t^2$ | $2\mathcal{O}_1+2\mathcal{O}_2+2\mathcal{O}_3$ | $\begin{pmatrix} z_1 & \frac{z_1}{2} \\ \frac{z_1}{2} & z_2 \end{pmatrix}$ |
| 5 | 4 | $\mathbb{Z}_4$ | $DV_1(1,1,8,1)$ | $t+t^4$ | $\mathcal{O}_3$ | $\begin{pmatrix} z_1 & 0 \\ 0 & i \end{pmatrix}$ |
| 6 | 6 | $W(\mathfrak{su}(3))=G(3,3,2)$ | $DV'_3(1,3)$ | $t^2+t^3$ | $\mathcal{O}_1+\mathcal{O}_2+\mathcal{O}_3$ | $\begin{pmatrix} z_1 & \frac{z_1}{2} \\ \frac{z_1}{2} & z_1+1 \end{pmatrix}$ |
| 7 | 6 | $\mathbb{Z}_2\times\mathbb{Z}_3$ | $DV_1(1,1,12,5)$ | $t^2+t^3$ | $\mathcal{O}_1+\mathcal{O}_2+2\mathcal{O}_3$ | $\begin{pmatrix} z_1 & 0 \\ 0 & e^{2\pi i/3} \end{pmatrix}$ |
| 8 | 6 | $\mathbb{Z}_6$ | $DV_1(1,1,12,1)$ | $t+t^6$ | $\mathcal{O}_3$ | $\begin{pmatrix} z_1 & 0 \\ 0 & e^{2\pi i/3} \end{pmatrix}$ |
| 9 | 8 | $W(\mathfrak{so}(5))=G(4,4,2)$ | $DV_4(1,1)$ | $t^2+t^4$ | $\mathcal{O}_1+\mathcal{O}_2+\mathcal{O}_3$ | $\begin{pmatrix} 2z_3 & z_3 \\ z_3 & \frac{z_3^2-1}{2z_3} \end{pmatrix}$ |
| 10 | 8 | $\mathbb{Z}_2\times\mathbb{Z}_4$ | $DV_1(2,2,4,1)$ | $t^2+t^4$ | $\mathcal{O}_1+\mathcal{O}_2+2\mathcal{O}_3$ | $\begin{pmatrix} z_1 & 0 \\ 0 & i \end{pmatrix}$ |
| 11 | 9 | $\mathbb{Z}_3\times\mathbb{Z}_3$ | $DV_1(3,3,2,1)$ | $2t^3$ | $2\mathcal{O}_3$ | $\begin{pmatrix} e^{2\pi i/3} & 0 \\ 0 & e^{2\pi i/3} \end{pmatrix}$ |
| 12 | 12 | $W(G_2)=G(6,6,2)$ | $DV_3(1,3)$ | $t^2+t^6$ | $\mathcal{O}_1+\mathcal{O}_2+\mathcal{O}_3$ | $\begin{pmatrix} z_1 & \frac{z_1}{2} \\ \frac{z_1}{2} & z_1+1 \end{pmatrix}$ |
| 13 | 12 | $\mathbb{Z}_2\times\mathbb{Z}_6$ | $DV_1(2,2,6,1)$ | $t^2+t^6$ | $\mathcal{O}_1+\mathcal{O}_2+2\mathcal{O}_3$ | $\begin{pmatrix} z_1 & 0 \\ 0 & e^{2\pi i/3} \end{pmatrix}$ |
| 14 | 12 | $\mathbb{Z}_3\times\mathbb{Z}_4$ | $DV_1(1,1,24,7)$ | $t^3+t^4$ | $2\mathcal{O}_3$ | $\begin{pmatrix} e^{2\pi i/3} & 0 \\ 0 & i \end{pmatrix}$ |
| 15 | 16 | $\mathbb{Z}_4\times\mathbb{Z}_4$ | $DV_1(4,4,2,1)$ | $2t^4$ | $2\mathcal{O}_3$ | $\begin{pmatrix} i & 0 \\ 0 & i \end{pmatrix}$ |
| 16 | 16 | $W(\mathfrak{so}(5))\rtimes\mathbb{Z}_2=G(4,2,2)$ | $DV_3(2,2)$ | $2t^4$ | $\mathcal{O}_3$ | $\begin{pmatrix} i & 0 \\ 0 & i \end{pmatrix}$ |
| 17 | 18 | $\mathbb{Z}_3\times\mathbb{Z}_6$ | $DV_1(3,3,4,1)$ | $t^3+t^6$ | $2\mathcal{O}_3$ | $\begin{pmatrix} e^{2\pi i/3} & 0 \\ 0 & e^{2\pi i/3} \end{pmatrix}$ |
| 18 | 18 | $W(\mathfrak{su}(3))\rtimes\mathbb{Z}_3=G(3,1,2)$ | $DV'_3(3,3)$ | $t^3+t^6$ | $\mathcal{O}_3$ | $\begin{pmatrix} e^{2\pi i/3} & 0 \\ 0 & e^{2\pi i/3} \end{pmatrix}$ |
| 19 | 24 | $G(6,3,2)$ | $DV_4(1,3)$ | $t^4+t^6$ | $\mathcal{O}_3$ | $\frac{1}{\sqrt{3}}\begin{pmatrix} 2i & i \\ i & 2i \end{pmatrix}$ |
| 20 | 24 | $\mathbb{Z}_4\times\mathbb{Z}_6$ | $DV_1(2,2,12,5)$ | $t^4+t^6$ | $2\mathcal{O}_3$ | $\begin{pmatrix} e^{2\pi i/3} & 0 \\ 0 & i \end{pmatrix}$ |
| 21 | 32 | $G(4,1,2)$ | $DV_4(2,2)=DV_3(2,4)$ | $t^4+t^8$ | $\mathcal{O}_3$ | $\begin{pmatrix} i & 0 \\ 0 & i \end{pmatrix}$ |
| 22 | 36 | $\mathbb{Z}_6\times\mathbb{Z}_6$ | $DV_1(6,6,2,1)$ | $2t^6$ | $2\mathcal{O}_3$ | $\begin{pmatrix} e^{2\pi i/3} & 0 \\ 0 & e^{2\pi i/3} \end{pmatrix}$ |
| 23 | 36 | $W(\mathfrak{su}(3))\rtimes\mathbb{Z}_6=G(6,2,2)$ | $DV_3(3,3)$ | $2t^6$ | $\mathcal{O}_3$ | $\begin{pmatrix} e^{2\pi i/3} & 0 \\ 0 & e^{2\pi i/3} \end{pmatrix}$ |
| 24 | 48 | $\mathrm{ST}_{12}^{\spadesuit}$ | $DV_8(1)$ | $t^6+t^8$ | $\mathcal{O}_3$ | $\frac{1}{3}\begin{pmatrix} 1+2i\sqrt{2} & -1+i\sqrt{2} \\ -1+i\sqrt{2} & 1+2i\sqrt{2} \end{pmatrix}$ |
| 25 | 72 | $G(6,1,2)$ | $DV_4(3,3)$ | $t^6+t^{12}$ | $\mathcal{O}_3$ | $\begin{pmatrix} e^{2\pi i/3} & 0 \\ 0 & e^{2\pi i/3} \end{pmatrix}$ |

$\spadesuit$ = $\mathrm{ST}_{12}$ is isomorphic to the binary octahedral group.

| Caption | Product of rank-1 theories | Known discrete gauging or S-folds |
|---|---|---|
| | Simple $\mathcal{N}=4$ theories | Theories with no known realization |

- Table 2 consists of the groups where $\mathrm{PLog}_\Gamma(t)$ is a polynomial with positive coefficients. In that case, the ring of invariants is freely generated. Since we only consider rank two geometries, $\mathrm{PLog}_\Gamma(t)$ takes the form $t^{d_1} + t^{d_2}$ where $d_1$ and $d_2$ are two positive integers (which can be equal). This means that the Coulomb branch is freely generated by two operators of dimensions $d_1$ and $d_2$.

- Table 3 consists of the groups where $\mathrm{PLog}_\Gamma(t)$ is a polynomial with one negative coefficient after some positive coefficients. For rank two geometries, in this case $\mathrm{PLog}_\Gamma(t)$

Table 3: List of geometries which are complete intersections as complex varieties. "Quaternion" stands for the order 8 quaternion group; $\text{Dic}_n$ is the order $4n$ dicyclic group; $\text{SD}_{16}$ is the semi-dihedral group of order 16; $M_n(2)$ is the order $2^n$ modular maximal-cyclic group ($M_4(2)$ is sometimes called $M_{16}$). See equations (79), (80) and (81) for an explicit definition. The meaning of the colors is the same as in Table 1, and in addition we painted in red the geometries that are excluded in Section 4.

| # | $|\Gamma|$ | $\Gamma$ | Du Val class | $\text{PLog}_\Gamma(t)$ | $\mathcal{O}(t^2)\,\text{PLog}_{\mathcal{M}_\Gamma}$ | $\text{Fix}(\Gamma)$ |
|---|---|---|---|---|---|---|
| 26 | 2 | $\mathbb{Z}_2$ | $\text{DV}_1(2,2,1,0)$ | $3t^2-t^4$ | $3\mathcal{O}_1+3\mathcal{O}_2+4\mathcal{O}_3+\mathcal{O}_4$ | $\begin{pmatrix} z_1 & z_3 \\ z_3 & z_2 \end{pmatrix}$ |
| 27 | 3 | $\mathbb{Z}_3$ | $\text{DV}_1(1,3,2,1)$ | $t^2+2t^3-t^6$ | $\mathcal{O}_1+\mathcal{O}_2+2\mathcal{O}_3+\mathcal{O}_4$ | $\begin{pmatrix} z_1 & \frac{z_1}{2} \\ \frac{z_1}{2} & z_1+1 \end{pmatrix}$ |
| 28 | 4 | $\mathbb{Z}_4$ | $\text{DV}_1(2,4,1,0)$ | $t^2+2t^4-t^8$ | $\mathcal{O}_1+\mathcal{O}_2+2\mathcal{O}_3+\mathcal{O}_4$ | $\begin{pmatrix} z_1 & -\frac{1}{2} \\ -\frac{1}{2} & z_1+1 \end{pmatrix}$ |
| 29 | 4 | $\mathbb{Z}_4$ | $\text{DV}_1(1,1,8,3)$ | $t^2+t^3+t^4-t^6$ | $\mathcal{O}_1+\mathcal{O}_2+2\mathcal{O}_3$ | $\begin{pmatrix} z_1 & 0 \\ 0 & i \end{pmatrix}$ |
| 30 | 6 | $\mathbb{Z}_6$ | $\text{DV}_1(2,6,1,0)$ | $t^2+2t^6-t^{12}$ | $\mathcal{O}_1+\mathcal{O}_2+2\mathcal{O}_3+\mathcal{O}_4$ | $\begin{pmatrix} z_1 & \frac{z_1}{2} \\ \frac{z_1}{2} & z_1+1 \end{pmatrix}$ |
| 31 | 6 | $\mathbb{Z}_6$ | $\text{DV}_1(2,2,3,1)$ | $t^2+t^4+t^6-t^8$ | $\mathcal{O}_1+\mathcal{O}_2+2\mathcal{O}_3$ | $\begin{pmatrix} z_1 & 0 \\ 0 & e^{2\pi i/3} \end{pmatrix}$ |
| 32 | 6 | $\mathbb{Z}_6$ | $\text{DV}_1(3,1,4,1)$ | $2t^3+t^6-t^9$ | $2\mathcal{O}_3$ | $\begin{pmatrix} e^{2\pi i/3} & 0 \\ 0 & e^{2\pi i/3} \end{pmatrix}$ |
| 33 | 8 | $\mathbb{Z}_2\times\mathbb{Z}_4$ | $\text{DV}_1(4,4,1,0)$ | $3t^4-t^8$ | $2\mathcal{O}_3$ | $\begin{pmatrix} i & 0 \\ 0 & i \end{pmatrix}$ |
| 34 | 8 | Quaternion | $\text{DV}_2(1,2)=\text{DV}_3(1,2)$ | $2t^4+t^6-t^{12}$ | $\mathcal{O}_3+\mathcal{O}_4$ | $\begin{pmatrix} i & 0 \\ 0 & i \end{pmatrix}$ |
| 35 | 12 | $\mathbb{Z}_2\times\mathbb{Z}_6$ | $\text{DV}_1(2,6,2,1)$ | $t^4+2t^6-t^{12}$ | $2\mathcal{O}_3$ | $\frac{1}{\sqrt{3}}\begin{pmatrix} 2i & i \\ i & 2i \end{pmatrix}$ |
| 36 | 12 | $\text{Dic}_3$ | $\text{DV}_2(1,3)$ | $t^4+t^6+t^8-t^{16}$ | $\mathcal{O}_3+\mathcal{O}_4$ | $\frac{1}{\sqrt{3}}\begin{pmatrix} 2i & i \\ i & 2i \end{pmatrix}$ |
| 37 | 12 | $\mathbb{Z}_{12}$ | $\text{DV}_1(1,1,24,5)$ | $t^4+t^5+t^6-t^{10}$ | $2\mathcal{O}_3$ | $\begin{pmatrix} e^{2\pi i/3} & 0 \\ 0 & i \end{pmatrix}$ |
| 38 | 16 | $\text{SD}_{16}$ | $\text{DV}_4(1,2)$ | $t^4+t^6+t^8-t^{12}$ | $\mathcal{O}_3$ | $\frac{1}{3}\begin{pmatrix} 1+2i\sqrt{2} & -1+i\sqrt{2} \\ -1+i\sqrt{2} & 1+2i\sqrt{2} \end{pmatrix}$ |
| 39 | 16 | $M_4(2)$ | $\text{DV}_4(2,1)$ | $t^4+2t^8-t^{16}$ | $\mathcal{O}_3$ | $\begin{pmatrix} i & 0 \\ 0 & i \end{pmatrix}$ |
| 40 | 18 | $\mathbb{Z}_3\times\mathbb{Z}_6$ | $\text{DV}_1(6,6,1,0)$ | $3t^6-t^{12}$ | $2\mathcal{O}_3$ | $\begin{pmatrix} e^{2\pi i/3} & 0 \\ 0 & e^{2\pi i/3} \end{pmatrix}$ |
| 41 | 24 | $\text{SL}(2,3)^{\clubsuit}$ | $\text{DV}_5(1)$ | $t^6+t^8+t^{12}-t^{24}$ | $\mathcal{O}_3+\mathcal{O}_4$ | $\frac{1}{3}\begin{pmatrix} 1+2i\sqrt{2} & -1+i\sqrt{2} \\ -1+i\sqrt{2} & 1+2i\sqrt{2} \end{pmatrix}$ |
| 42 | 24 | $W(\mathfrak{so}(5))\rtimes\mathbb{Z}_3$ | $\text{DV}_4(3,1)$ | $2t^6+t^{12}-t^{18}$ | $\mathcal{O}_3$ | $\begin{pmatrix} e^{2\pi i/3} & 0 \\ 0 & e^{2\pi i/3} \end{pmatrix}$ |
| 43 | 36 | $\text{Dic}_3\times\mathbb{Z}_3$ | $\text{DV}_2(3,3)$ | $t^6+2t^{12}-t^{24}$ | $\mathcal{O}_3$ | $\begin{pmatrix} e^{2\pi i/3} & 0 \\ 0 & e^{2\pi i/3} \end{pmatrix}$ |

$\clubsuit$ = $\text{SL}(2,3)$ is isomorphic to the binary tetrahedral group.

| Caption | Discrete gauging of $U(1)^2\,\mathcal{N}=4$ | Known discrete gauging or S-folds |
|---|---|---|
| | Excluded geometries | Theories with no known realization |

takes the form $t^{d_1}+t^{d_2}+t^{d_3}-t^{d_4}$. This means the Coulomb branch is a complete intersection, generated by three operators of dimensions $d_1$, $d_2$ and $d_3$ which satisfy an algebraic relation at degree $d_4$.

- Table 4 lists the other groups, where $\text{PLog}_\Gamma(t)$ is an infinite series.

In those tables, the first column is a label that we use to identify the geometries. We found 53 geometries. The second column gives the cardinality of the group and the third column give an abstract group isomorphic to the corresponding finite U(2) subgroup. The fourth column gives the Du Val class, followed in the fifth column by the unrefined Hilbert series of the ring of invariants (given in the form of $\text{PLog}_\Gamma(t)$ in Tables 2 and 3 where the latter is a polynomial, and given as a rational function in Table 4). Finally, the last two columns give the order $t^2$ of the PLog of the refined Hilbert series (more below) and the set of $\text{Fix}(\Gamma)$ in parametric form, as described previously.

There is some arbitrariness in our choice of naming of the abstract groups appearing in column three. For instance in the case of cyclic groups, we have the isomorphism $\mathbb{Z}_p\times\mathbb{Z}_q=\mathbb{Z}_{pq}$ if $p$ and $q$ are mutually prime. For some non-abelian groups of low order, different names can

Table 4: List of geometries whose holomorphic coordinate rings are neither freely generated nor complete intersections. Note that all the groups are abelian. Since the PLog of the Hilbert series have no simple expression, we tabulate the Hilbert series themselves. For an explanation of the coloring code, see Section 4.1.

| # | $|\Gamma|$ | $\Gamma$ | Du Val class | $H_\Gamma(t)$ | $\mathcal{O}(t^2)\,\mathrm{PLog}_{\mathcal{M}_\Gamma}$ | Fix($\Gamma$) |
|---|---|---|---|---|---|---|
| 44 | 3 | $\mathbb{Z}_3$ | $DV_1(3,1,2,1)$ | $\frac{2t^3+1}{(1-t^3)^2}$ | $4\mathcal{O}_3$ | $\begin{pmatrix} e^{2\pi i/3} & 0 \\ 0 & e^{2\pi i/3} \end{pmatrix}$ |
| 45 | 4 | $\mathbb{Z}_4$ | $DV_1(4,2,1,0)$ | $\frac{3t^4+1}{(1-t^4)^2}$ | $4\mathcal{O}_3$ | $\begin{pmatrix} i & 0 \\ 0 & i \end{pmatrix}$ |
| 46 | 5 | $\mathbb{Z}_5$ | $DV_1(1,1,10,3)$ | $\frac{t^7+t^6+t^4+t^3+1}{(1-t^5)^2}$ | $2\mathcal{O}_3$ | $\begin{pmatrix} e^{2\pi i/5} & \frac{1}{2}i\left(\sqrt{5-2\sqrt5}+i\right) \\ \frac{1}{2}i\left(\sqrt{5-2\sqrt5}+i\right) & e^{3\pi i/5} \end{pmatrix}$ |
| 47 | 6 | $\mathbb{Z}_6$ | $DV_1(1,3,4,1)$ | $\frac{t^5+t^4+1}{(1-t^3)(1-t^6)}$ | $2\mathcal{O}_3$ | $\frac{1}{\sqrt3}\begin{pmatrix} 2i & i \\ i & 2i \end{pmatrix}$ |
| 48 | 6 | $\mathbb{Z}_6$ | $DV_1(6,2,1,0)$ | $\frac{5t^6+1}{(1-t^6)^2}$ | $4\mathcal{O}_3$ | $\begin{pmatrix} e^{2\pi i/3} & 0 \\ 0 & e^{2\pi i/3} \end{pmatrix}$ |
| 49 | 8 | $\mathbb{Z}_8$ | $DV_1(4,2,2,1)$ | $\frac{2t^8+t^4+1}{(1-t^4)(1-t^8)}$ | $2\mathcal{O}_3$ | $\begin{pmatrix} i & 0 \\ 0 & i \end{pmatrix}$ |
| 50 | 8 | $\mathbb{Z}_8$ | $DV_1(2,4,2,1)$ | $\frac{t^8+2t^6+1}{(1-t^4)(1-t^8)}$ | $2\mathcal{O}_3$ | $\frac{1}{3}\begin{pmatrix} 1+2i\sqrt2 & -1+i\sqrt2 \\ -1+i\sqrt2 & 1+2i\sqrt2 \end{pmatrix}$ |
| 51 | 10 | $\mathbb{Z}_{10}$ | $DV_1(2,2,5,2)$ | $\frac{t^{10}+t^8+t^6+1}{(1-t^4)(1-t^{10})}$ | $2\mathcal{O}_3$ | $\begin{pmatrix} e^{2\pi i/5} & \frac{1}{2}i\left(\sqrt{5-2\sqrt5}+i\right) \\ \frac{1}{2}i\left(\sqrt{5-2\sqrt5}+i\right) & e^{3\pi i/5} \end{pmatrix}$ |
| 52 | 12 | $\mathbb{Z}_2\times\mathbb{Z}_6$ | $DV_1(6,2,2,1)$ | $\frac{2t^6+1}{(1-t^6)^2}$ | $2\mathcal{O}_3$ | $\begin{pmatrix} e^{2\pi i/3} & 0 \\ 0 & e^{2\pi i/3} \end{pmatrix}$ |
| 53 | 12 | $\mathbb{Z}_{12}$ | $DV_1(6,4,1,0)$ | $\frac{3t^{12}+2t^6+1}{(1-t^6)(1-t^{12})}$ | $2\mathcal{O}_3$ | $\begin{pmatrix} e^{2\pi i/3} & 0 \\ 0 & e^{2\pi i/3} \end{pmatrix}$ |

Caption    Excluded geometries

be used depending on the context; we tried to use the most common ones in the table. To help the reader, we recall the definition of the various finite groups that we used in the captions of the various tables, and, in appendix B, we give the definition of the $G(m,p,r)$ series of complex reflection groups.[8]

It is important to realize that the same abstract group can correspond to various distinct geometries, depending on the way it is embedded in U(2). We give below an example of this fact involving a complex reflection group. The Du Val label, on the contrary, entirely and unambiguously characterizes the geometry since it encodes not just the abstract group but also the equivalence class of its embedding in U(2). The Du Val classes are reviewed in appendix A.

## 3.4 Complex singularity structure

In section 2, we discussed at length the metric singularity structures of the orbifold geometries and their physical interpretation. Here we will spend a few words to describe, instead, the complex singularities of these spaces. The first point to emphasize is that the locus of complex singularities is generically a proper subvariety of the locus of metric singularities. The second point is that these kinds of singularities have clearly distinct physical interpretations: metric singularities occur where states charged under the low energy $\mathrm{U}(1)^r$ gauge group become massless, and signal the occurrence of non-trivial either interacting SCFT or IR free theories in the IR; complex singularities occur whenever the chiral ring of moduli space operators of the IR SCFT is not freely generated.

Let us first consider the possible dimensionality of the complex singularities. It is a well-known fact that rank-1 orbifolds do not develop complex structure singularities (this can be deduced, for instance, from Table 5 by inspection). As a consequence, we expect no co-dimension 1 complex singularities in a rank-$r$ orbifold. In the particular case of rank 2, this means that all

---

[8]We have used the standard definition for the groups $G(m,p,r)$; note that it differs from the non-standard notation used in [12], where the positions of $m$ and $r$ are exchanged.

complex singularities will be points, and by conformal invariance there can be only one such point, namely the origin. The question then boils down to, for each geometry, how singular (if at all) the origin is.

By definition, all the geometries presented in Table 2 are freely generated, which means that the orbifolds are isomorphic to $\mathbb{C}^2$ as complex algebraic varieties, and there are no complex singularities in those cases. On the other hand, every geometry listed in Tables 3 and 4 contain complex singularities. These can be characterized algebraically, using a few tools from invariant theory and commutative algebra, namely

- the averaging trick to generate invariant polynomials of a given degree under a finite group;

- the Molien formula (42) to compute the Hilbert series of the invariant ring of a finite matrix group;

- an algorithm to compute the Hilbert series of a polynomial ring defined by a homogeneous ideal.[9]

Using these tools, given a finite matrix group $G = \mu(\Gamma) \subset \mathrm{GL}(r, \mathbb{C})$ acting by left multiplication on $(z_1, \ldots, z_r) \in \mathbb{C}^r$ one proceeds as follows:[10]

1. Compute the Hilbert series $H_G(t)$ of the ring of invariants of $G$ using (42), and look at the lowest power $d > 0$ in its series expansion. This signals the fact that there exists an invariant polynomial of $G$ at degree $d$.

2. Apply the averaging trick on a basis of homogeneous polynomials in the $z_i$ of degree $d$ until a non-zero invariant $a_1 = P_1(z_1, \ldots, z_r)$ is produced. Compute the Hilbert series $H_1(t)$ of the ring $\mathbb{C}[z_1, \ldots, z_r, a_1]/I_1$, where $I_1$ is the ideal generated by $P_1$.

3. If the Hilbert series is equal to $H_G(t)$, we have the ring of invariants. Otherwise, repeat steps 1 and 2 starting with the difference $H_G(t) - H_1(t)$. This will generate a second invariant $a_2 = P_2(z_1, \ldots, z_r)$ and the Hilbert series $H_2(t)$ of $\mathbb{C}[z_1, \ldots, z_r, a_1, a_2]/I_2$, where $I_2$ is the ideal generated by $P_1$ and $P_2$. Iterate these steps until for some $m$, $H_G(t) = H_m(t)$.

4. We now know that the ring of invariants is described by $\mathbb{C}[z_1, \ldots, z_r, a_1, \ldots, a_m]/I_m$. Compute the elimination ideal in which the $z_1, \ldots, z_r$ have been eliminated, and choose a system of generators, $f_i$, $i = 1, \ldots, p$, of this ideal. These correspond to algebraic relations satisfied by the invariants $a_1, \ldots, a_m$. In other words, these describe the Coulomb branch as a complex algebraic variety.

5. Compute the singular locus of this algebraic variety by computing the rank of the $p \times m$ matrix of derivatives $\partial f_i / \partial a_k$. To check whether they vanish on the variety, one can compute a Gröbner basis associated to the $f_i$ and use Euclid's algorithm to reduce all the minors with respect to this basis. We compute on minors of decreasing order until we find the order, $s$, at which they do not vanish, meaning that the variety has dimension $r - s$. The subvariety of complex singularities is then given by the vanishing of all the minors of order $s$.

Many explicit examples are given in [14] but let's work out here how things work in one example.

---

[9]This algorithm can be time consuming, as it involves a Gröbner basis computation.
[10]See also appendix A of [13] for the same algorithm applied to Lie groups.

**Example: geometry number 51.** For entry 51, the geometry is $DV_1(2,2,5,2)$. The associated group is cyclic of order 10, generated for instance by the element corresponding to $x = 8$ and $y = 9$ in (84). This means the action on $\mathbb{C}^2$ is given by

$$(z_1, z_2) \sim (\omega^3 z_1, \omega z_2), \tag{45}$$

where $\omega = e^{-2\pi i/10}$. The Hilbert series of the ring of invariants, computed using the Molien formula, is

$$\frac{1 + t^6 + t^8 + t^{10}}{(1 - t^4)(1 - t^{10})}. \tag{46}$$

Steps 2 and 3 in the algorithm above produce a set of 5 invariants

$$
\begin{aligned}
a_1 &= z_1^3 z_2 \\
a_2 &= z_1^2 z_2^4 \\
a_3 &= z_1 z_2^7 \\
a_4 &= z_1^{10} \\
a_5 &= z_2^{10}.
\end{aligned}
$$

It turns out that the ring of invariants is generated by two primary invariants of degrees 4 and 10, and three secondary invariants of degrees 6, 8 and 10, corresponding to the standard form (46). Using the elimination ideal, one finds a description of the algebraic variety as the set of zeroes of the polynomials

$$
\begin{aligned}
f_1 &= a_1 a_3 - a_2^2 & (47) \\
f_2 &= a_2 a_3 - a_1 a_5 & (48) \\
f_3 &= a_3^2 - a_2 a_5 & (49) \\
f_4 &= a_1^4 - a_2 a_4 & (50) \\
f_5 &= a_4 a_3 - a_1^3 a_2 & (51) \\
f_6 &= a_1^3 a_3 - a_4 a_5. & (52)
\end{aligned}
$$

To cross-check the validity of the result, the Hilbert series of the ideal generated by these 6 polynomials should correspond to the wanted Hilbert series (46), divided by $(1-t)^2$ to account for the two eliminated generators $z_1, z_2$.

Now we want to find the singular locus. We look at the matrix of the partial derivatives $\partial f_i/\partial x$ for $i = 1, \dots, 6$ and $x = a, b, c, d, e$, and check that on the variety it generically has rank 3. The singular points correspond to the points where the rank drops, which is given by the vanishing of all the $3 \times 3$ minors. Due to the large number of order 3 minors, it is easy to find a necessary and sufficient condition for their joint vanishing. One finds that the only singular point is the origin $a = b = c = d = e = 0$.

## 3.5 Remarks

Before delving deeper in understanding the physics associated to the geometries associated to the groups discussed in this section, let us point out various subtleties from the mathematical point of view.

- The freely generated geometries must correspond to (not necessarily irreducible) crystallographic complex reflection groups (CRGs) [10]. These groups have been classified [34]. At rank two, the CRGs are either of the form $\mathbb{Z}_{d_1} \times \mathbb{Z}_{d_2}$ for $d_i \in \{1, 2, 3, 4, 6\}$ (reducible action), or of the form $G(m, p, 2)$, or one of the 19 exceptional rank two

CRGs, labeled by their Shephard–Todd name, $ST_i$ for $4 \leq i \leq 22$. It is a consistency check that all the Du Val groups appearing in Table 2 indeed correspond to one of these crystallographic CRGs.

- Weyl groups are a special case. In fact they are the only crystallographic real reflection groups. This has two consequences. By being a reflection group, the CB coordinate ring of the resulting geometry is freely generated and the CB is isomorphic to $\mathbb{C}^r$, $r$ being the rank. Secondly, by being real, the Weyl group action preserves a symmetric bilinear form which implies that one of their CB generator has always scaling dimension 2. This in turn, because of $\mathcal{N} = 3$ supersymmetry, implies that the theory has an exactly marginal operator and an extra supercharge. Thus we recover the expected result that Weyl groups define the moduli spaces of $\mathcal{N} = 4$ SCFTs.

- As we mentioned above, a given abstract group does not characterize the geometry if its action on $\mathbb{C}^2$ is not specified. This explains why many abstract groups that can be seen in some embeddings as CRGs do not give rise to freely generated geometries. This is the case of all the cyclic groups appearing in Tables 3 and 4. For a more striking example, geometry number 41 in Table 3 is an orbifold of $\mathbb{C}^2$ by the binary tetrahedral group, which is isomorphic as an abstract group to a complex reflection group, called $ST_4$ in the Shephard–Todd classification. In this case the $U(2)$ finite subgroup specified by the Du Val label is the order 24 group generated by the two matrices

$$\begin{pmatrix} 0 & i \\ i & 0 \end{pmatrix} \quad \text{and} \quad \frac{1}{2}\begin{pmatrix} 1+i & 1+i \\ -1+i & 1-i \end{pmatrix}. \tag{53}$$

None of these matrices is a complex reflection, since they both fix only the origin. [11]

- In Table 4, we have given the Hilbert series as a rational function of $t$ in the form

$$\frac{P(t)}{(1-t^{d_1})(1-t^{d_2})}, \tag{57}$$

where $P(t)$ is a polynomial with positive integer coefficients. When this is the case, we say that the Hilbert series is written in a *standard form*. One often interprets a Hilbert series of this form as saying that there are two primary generators of dimensions $d_1$ and $d_2$. However such a form is not unique in general. For instance, taking the Hilbert series for geometry number 51, we have

$$\frac{t^{10} + t^8 + t^6 + 1}{(1-t^4)(1-t^{10})} = \frac{t^{12} + t^{10} + 2t^8 + t^4 + 1}{(1-t^6)(1-t^{10})}. \tag{58}$$

This simply means that the denominators of the Hilbert series in Table 4 should not be interpreted as the degrees of two generators of the Coulomb branch. A deeper analysis is necessary in those cases, such as the one given above for entry 51. Thus the notion of a primary operator is neither uniquely defined nor particularly useful to characterize the physics of the corresponding geometry.

---

[11] The invariant ring $\mathbb{C}[z_1, z_2]^G$ is generated by the three invariant polynomials

$$a = z_1 z_2 \left( z_2^4 - z_1^4 \right) \tag{54}$$
$$b = z_1^8 + 14 z_2^4 z_1^4 + z_2^8 \tag{55}$$
$$c = z_1^{12} - 33 z_2^4 z_1^8 - 33 z_2^8 z_1^4 + z_2^{12} \tag{56}$$

of respective degrees 6, 8 and 12 which satisfy the relation $b^3 = 108 a^4 + c^2$ at degree 24, in agreement with the PLog given in the table.

- The ring of invariants of a finite group Γ in a given representation admits what is called a Hironaka decomposition (see [35] for a review), which identifies primary and secondary generators of the ring. A Hilbert series in standard form is associated to such a decomposition. However it should be emphasized that in addition to the non-uniqueness of the standard form discussed above, it can happen that no Hironaka decomposition of the ring of invariants corresponds to a given standard form. An example of such a situation is given in [36], section 2.1.

## 4 Constraints from the Hilbert series

The last section outlined how the classification of all TSK orbifold geometries is carried out, giving the entries of Tables 2, 3 and 4. In this section we will show that additional constraints on physically allowed $\mathcal{M}_\Gamma$ can be inferred by considering the Hilbert series of the entire moduli space of vacua and not only that associated to a single irreducible action on an $\mathcal{N} = 2$ CB section $\mathbb{C}^r$. The way that the special coordinates on $\mathbb{C}^{3r}$ transform under the action of the non-holomorphic $U(3)_R$ isometry of $\mathcal{M}_\Gamma$ carries non-trivial information about the operators whose vevs parametrize the moduli space and thus constrain the minimal operator content of a putative theory $\mathcal{T}_\mathcal{M}$ realizing the moduli space. In particular we are able to count the number of stress tensors of $\mathcal{T}_\mathcal{M}$ and in some cases the presence of higher spin currents which show the existence of a free sector in $\mathcal{T}_\mathcal{M}$. This in turn, as we will explain in detail below, can be turned into a set of constraints which further refine our set of admissible geometries captured by the color shading on the tables.

Let us first introduce some notation. Representations of $U(3)_R = SU(3)_R \times U(1)_R$ will be denoted $(R_1, R_2; r)$ where $(R_1, R_2)$ are the Dynkin labels of $SU(3)_R$ and $r$ is the $U(1)_R$ charge. A generic weight in this representation will be denoted similarly $(\lambda_1, \lambda_2; r)$.

As mentioned, the main object we use is the Hilbert series of the coordinate ring of the moduli space $\mathcal{M}_\Gamma$ (12) [14,32,33,37–39]. Since $\mathcal{M}_\Gamma$ carries a non-holomorphic $U(3)_R$ isometry, we can consider a refined version of the Hilbert series, given by the Molien formula (42) as

$$H_{\mathcal{M}_\Gamma}(t, v, \boldsymbol{u}) = \frac{1}{|\Gamma|} \sum_{g \in \Gamma} \frac{1}{\det\left(\mathbf{1} - t \cdot v \cdot u_1 \cdot \mu_\tau(g)\right)} \frac{1}{\det\left(\mathbf{1} - t \cdot v \frac{u_2}{u_1} \mu_\tau(g)\right)} \frac{1}{\det\left(\mathbf{1} - \frac{t}{v} u_2 \overline{\mu}_\tau(g)\right)}.$$
(59)

Let us now pause and discuss (59). First of all, the fact that the Hilbert series factorizes in three pieces is an immediate consequence of the fact that the group action on $\mathbb{C}^6$ is chosen to be a direct sum of three factors $\rho(g) = \mu_\tau(g) \oplus \mu_\tau(g) \oplus \overline{\mu}_\tau(g)$, each individually acting on a $\mathbb{C}^2$. To understand the choices of fugacities in (59), recall that the holomorphic coordinates on $\mathcal{M}_\Gamma$ are (11)

$$(z_i^1, z_i^2, z_{3i}) := (a_i^1, a_i^2, \overline{a}_{3i}), \qquad i = 1, \dots, 2,$$
(60)

where the $a_i^I$ are the scalar primaries of a free $\mathcal{N} = 3$ vector multiplet which transform in a fundamental $(1, 0; 1)$ of the $U(3)_R$ symmetry. A straightforward group theory calculation then shows that the $U(3)_R$ weights of the holomorphic coordinates above are

|         | $\lambda_1$ | $\lambda_2$ | $r$ |
|---------|-------------|-------------|-----|
| $z_i^1$ | 1           | 0           | 1   |
| $z_i^2$ | −1          | 1           | 1   |
| $z_{3i}$ | 0          | 1           | −1  |

(61)

The powers of the $(u_1, u_2)$ fugacities in (59) are the $SU(3)_R$ weights while that of $v$ is the $U(1)_R$ charge. Note the effect of the complex conjugation in (11): while the character of the

fundamental representation of $SU(3)_R$ is $u_1 + \frac{u_2}{u_1} + \frac{1}{u_2}$ (with $u_i$, $i = 1, 2$, complex numbers of unit norm, so that $\bar{u}_i = \frac{1}{u_i}$), the weights that appear in (59) are instead $u_1$, $\frac{u_2}{u_1}$ and $u_2$. The parameter $t$ is redundant but we keep it for convenience in tracking the scaling dimension of various terms in the Hilbert series. Is worthwhile to also introduce the Hilbert series of the coordinate ring of the $\mathcal{N} = 2$ CB and HB slices of $\mathcal{M}_\Gamma$ which are defined in (13) and (15). It is straightforward to then reduce (59) to obtain

$$H_{\mathcal{C}_\Gamma}(t, v, \boldsymbol{u}) = \frac{1}{|\Gamma|} \sum_{g \in \Gamma} \frac{1}{\det\left(\boldsymbol{1} - t \cdot v \cdot u_1 \cdot \mu_\tau(g)\right)}, \tag{62}$$

$$H_{\mathcal{H}_\Gamma}(t, v, \boldsymbol{u}) = \frac{1}{|\Gamma|} \sum_{g \in \Gamma} \frac{1}{\det\left(\boldsymbol{1} - t \cdot v \frac{u_2}{u_1} \mu_\tau(g)\right)} \frac{1}{\det\left(\boldsymbol{1} - \frac{t}{v} u_2 \bar{\mu}_\tau(g)\right)}. \tag{63}$$

The definition of the PLog in this case is a straightfoward generalization of (43):

$$\text{PLog}_{\mathcal{M}_\Gamma/\mathcal{H}_\Gamma/\mathcal{C}_\Gamma}(t, v, \boldsymbol{u}) = \sum_{k=1}^{\infty} \frac{\mu(k)}{k} \log\left(H_{\mathcal{M}_\Gamma/\mathcal{H}_\Gamma/\mathcal{C}_\Gamma}(t^k, v^k, \boldsymbol{u}^k)\right), \tag{64}$$

where $\boldsymbol{u}^k := (u_1^k, u_2^k)$ and $\mu(k)$ is defined in (44).

Now let's discuss the operators which can acquire a vev parametrizing an $\mathcal{N} = 3$ moduli space of vacua and thus can be tracked by the PLog of the Hilbert series defined above. As discussed at length in [7], these are operators which are scalars, saturate a bound relating their scaling dimensions and their R-charges and they are chiral in the chosen complex structure, that is they are annihilated by the supercharge $\overline{Q}^{(0,1;1)}$.[12] $\mathcal{N} = 3$ superconformal invariance constrains how these operators can appear. Here we will summarize the main ingredients needed, for a more in-depth discussion of $\mathcal{N} = 3$ chiral rings see [7–9].

$\mathcal{N} = 3$ chiral multiplets or anti-chiral multiplets are those of types $X\overline{B}_1$ or their conjugates in table 18 of [40].[13] Complex conjugation acts on the quantum numbers of a representation as follows:

$$\mathbf{R} = (R_1, R_2) \mapsto \overline{\mathbf{R}} = (R_2, R_1),$$
$$\boldsymbol{\lambda} = (\lambda_1, \lambda_2) \mapsto -\boldsymbol{\lambda} = (-\lambda_1, -\lambda_2), \tag{65}$$

where we have also shown its action on $SU(3)_R$ weights. A systematic discussion of all operators which can be counted by the PLog of (59) is outside the scope of this paper; we only mention that an analysis of the PLog of the Hilbert series of $\mathcal{M}_\Gamma$ can be enough to reconstruct the VOA associated to $\mathcal{T}_\mathcal{M}$ [41]. Here we will focus on a few special operators.

The special multiplets we are interested in are the stress tensor multiplet and those which, along with operators which parametrize $\mathcal{N} = 3$ moduli space, also contain higher spin currents. These are, in the nomenclature of [40],[14]

- $B_1\overline{B}_1[0; 0]_2^{(1,1;0)}$, which contains the stress tensor.

- The only two multiplets with both a chiral ring operator and higher spin currents are

    - $A_2\overline{B}_1[0; 0]_2^{(1,0;8)}$ and its complex conjugate $B_1\overline{A}_2[0; 0]_2^{(0,1;-8)}$,

---

[12]Here the superscript gives the $U(3)_R$ weights of the operator, and not the Dynkin labels of a representation. Thus $\overline{Q}^{(0,1;1)}$ refers to the highest weight component of the representation and not the whole triplet of supercharges transforming in the antifundamental representation $(0, 1; 1)$.

[13]It is useful to present a dictionary between the nomenclature of [40] and that of [9]: $B_1\overline{B}_1 = \widehat{\mathcal{B}}_{[R_1,R_2]}$, $A_\ell\overline{B}_1 = \overline{\mathcal{D}}_{[R_1,R_2],j}$, $L\overline{B}_1 = \overline{\mathcal{B}}_{[R_1,R_2],R_3,j}$.

[14]The superscripts indicate the $U(3)_R$ Dynkin labels, the subscript the scaling dimension of the superconformal primary and between the square brackets are the Lorentz spins.

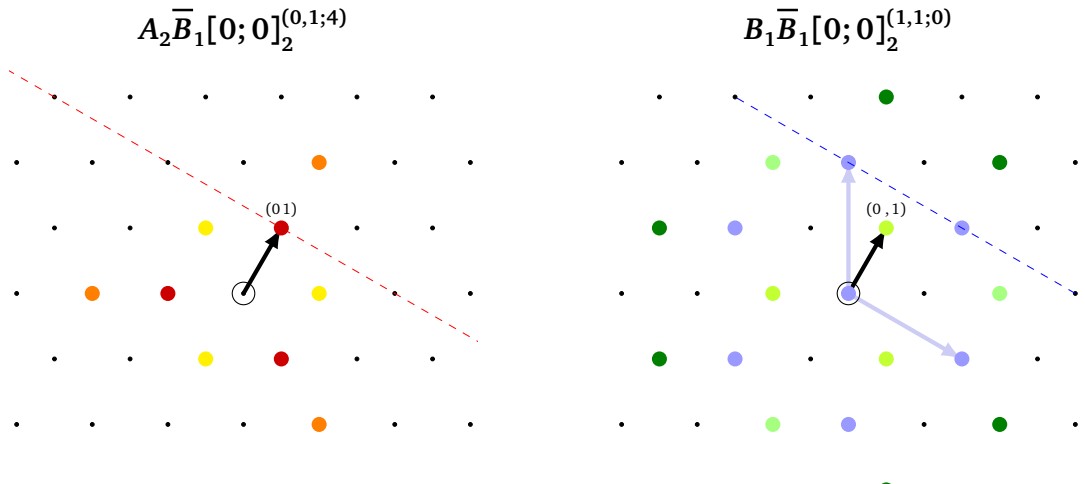

Figure 1: SU(3) weight lattice, with the vector showing our choice of the weight of the supercharge, $\overline{Q}^{(0,1;1)}$ defining the chiral ring. (a) Red dots are the weights of the $(0,1)$ representation that correspond to $A_2\overline{B}_1[0;0]_2^{(0,1;4)}$. The product $(0,1)\otimes(0,1)$ decomposes in $(0,2)$ (in orange and yellow) which contains the null states, and $(1,0)$ (represented in yellow) which are non-null states. The components of a chiral multiplet in the $\overline{\mathbf{3}}$ annihilated by $\overline{Q}^{(0,1;1)}$ lie on the dashed line. (b) Blue dots are the $(1,1)$ weights corresponding to $B_1\overline{B}_1[0;0]_2^{(1,1;0)}$. The product $(1,1)\otimes(0,1)$ decomposes into the null states $(1,2)$ (all green dots, dark and light) and the non-null states $(2,0)$ and $(0,1)$ (lighter shades of green). The components of a chiral multiplet in the $(1,1)$ annihilated by $\overline{Q}^{(0,1;1)}$ lie on the dashed line. The light blue arrows show the choice of simple roots with respect to which our Dynkin labels are defined.

 – $A_2\overline{B}_1[0;0]_2^{(0,1;4)}$ and its complex conjugate $B_1\overline{A}_2[0;0]_2^{(1,0;-4)}$.

The $[0;0]$ indicates that Lorentz spin of the superconformal primary is a scalar in all of these cases and it is in fact the operator contributing to the chiral ring. Identifying exactly which components of the superconformal primary are annihilated by the $\overline{Q}^{(0,1;1)}$ supercharge is a straightforward group theory exercise which we will now review; for more details see [7].

Consider a superconformal primary in the $(R_1,R_2;r)$ irreducible representation (irrep) of $U(3)_R$. Acting by $\overline{Q}$ we obtain operators transforming in the $((R_1,R_2)\otimes(0,1);r+1)$ which is in general a reducible representation. The null states lie in the $(R_1,R_2+1;r+1)$ [40], which is only one of the possibly many irreps into which the tensor product decomposes. A component of the $(R_1,R_2;r)$ irrep with weight $(\lambda_1,\lambda_2;r)$, is mapped by the top component $\overline{Q}^{(0,1;1)}$ to a state with weight $(\lambda_1,\lambda_2+1;r+1)$. The null representation always contains a component with such a weight, but that is not enough to assert that $(\lambda_1,\lambda_2;r)$ is annihilated by $\overline{Q}^{(0,1)}$ since $(\lambda_1,\lambda_2+1;r+1)$ might also appear as a weight of a non-null representation in the decomposition of $((R_1,R_2)\otimes(0,1);r+1)$. We conclude that $(\lambda_1,\lambda_2;r)$ is null if and only if $(\lambda_1,\lambda_2+1;r+1)$ does not appear in any non-null irreps of $((R_1,R_2)\otimes(0,1);r+1)$. This can be understood very easily by drawing the weight diagrams; see Figure 1. (We did not draw the weight diagrams corresponding to $A_2\overline{B}_1[0;0]_2^{(1,0;8)}$ or its conjugate since neither ever appear in the orbifold geometries analyzed here.)

In order to be able to isolate the contributions from the stress tensor multiplet and those containing higher spin currents, which are all dimension-2 operators, we need to also discussed other $\mathcal{N}=3$ chiral multiplet which contribute a chiral ring operator of scaling di-

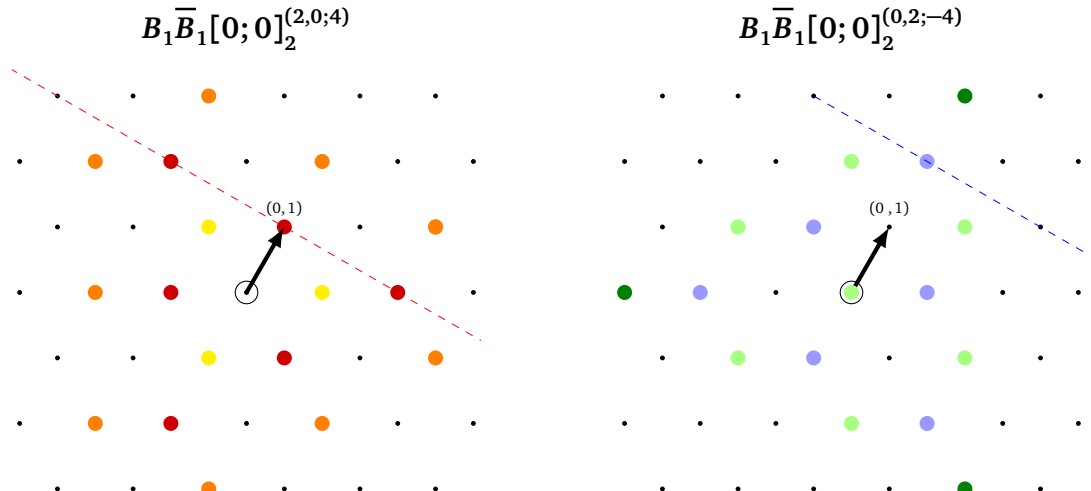

Figure 2: We use the same conventions as in Figure 1. (a) The product of the $(2,0)$ (in red) with $(0,1)$ gives the null states in the $(2,1)$ (in pink and yellow) and the non-null states $(1,0)$ (in yellow). (b) The product of the $(0,2)$ (in blue) with $(0,1)$ gives the null states in the $(0,3)$ (in green) and the non-null states in the $(1,1)$ (in light green). In both cases the component of the chiral multiplets annihilated by $\overline{Q}^{(0,1;1)}$ lie on the dashed line.

mension 2. Luckily the only such multiplets are $B_1\overline{B}_1[0;0]_2^{(0,2;-4)}$ and $B_1\overline{B}_1[0;0]_2^{(2,0;4)}$. These multiplets are very special as they contain an exactly marginal deformation operator, an extra supersymmetry-current multiplet, and an $\mathcal{N} = 2$ Coulomb branch operator of scaling dimension 2. Consistent with what we said in previous section, if these multiplets are present than the theory has an enlarged $\mathcal{N} = 4$ supersymmetry. The components of the superconformal primary of these operators annihilated by $\overline{Q}^{(0,1;1)}$ are depicted in Figure 2.

The occurrence of chiral primaries from these multiplets are easily extracted from the PLog (59). Based on Figures 1 and 2, we identify four types of contributions at order $t^2$ of the PLog, each representing a different $\mathcal{N} = 3$ multiplet. We call them $\mathcal{O}_1$, $\mathcal{O}_2$, $\mathcal{O}_3$ and $\mathcal{O}_4$ and they correspond to the following "characters":[15]

$$
\begin{aligned}
B_1\overline{B}_1^{(0,2;-4)} &\equiv \mathcal{O}_1 = \frac{u_2^2}{v^2}, \\
B_1\overline{B}_1^{(2,0;4)} &\equiv \mathcal{O}_2 = v^2\left(\frac{u_2^2}{u_1^2} + u_1^2 + u_2\right), \\
B_1\overline{B}_1^{(1,1;0)} &\equiv \mathcal{O}_3 = \frac{u_2^2}{u_1} + u_1 u_2, \\
A_2\overline{B}_1^{(0,1;4)} &\equiv \mathcal{O}_4 = u_2 v^2.
\end{aligned}
\tag{66}
$$

The result of computing the PLog (59) are reported in Tables 2, 3 and 4. Since all the special operators we have discussed have scaling dimension two, we only show the character decomposition of the order $t^2$ terms of the PLog. The generic form of such a term is $c_1\mathcal{O}_1 + c_2\mathcal{O}_2 + c_3\mathcal{O}_3 + c_4\mathcal{O}_4$ where the $c_i$ are interpreted as follows:

- $c_1 = c_2$ counts the number of extra supersymmetry current multiplets, which each also contain an exactly marginal operator and a $\mathcal{N} = 2$ CB operator of scaling dimension 2.

---

[15]We adopt the convention that the power of $v$ is half the $U(1)_R$ charge.

If $c_2 \neq 0$ there is a supersymmetry enhancement and $\mathcal{T}_\mathcal{M}$ is an $\mathcal{N} = 4$ theory.

- $c_3$ counts the number of stress-tensor multiplets. If $c_3 > 1$ then $\mathcal{T}_\mathcal{M}$ theory is a product theory.

- $c_4$ counts the number of higher-spin multiplets. If $c_4 \neq 0$, then $\mathcal{T}_\mathcal{M}$ has a free sector.

To get a better feeling for the information which can be extracted by analyzing the $\mathcal{O}(t^2)$ terms of $\mathrm{PLog}_{\mathcal{M}_\Gamma}$, let's look in detail at a few examples.

**Example** Focus first on entry #6 of Table 2. This entry describes the moduli space of vacua of the $\mathfrak{su}(3)$ $\mathcal{N} = 4$ theory. Because of the supersymmetry enhancement to $\mathcal{N} = 4$ we expect $c_1 = c_2 \neq 0$. Moreover since the theory has a single exactly marginal operator, we expect $c_1 = c_2 = 1$. We also expect a single stress-tensor, that is $c_3 = 1$ and no higher spin currents to be present as the theory is interacting, thus $c_4 = 0$. So these well-known facts about the $\mathfrak{su}(3)$ $\mathcal{N} = 4$ theory lead us to conclude that the 6th column of entry #6 of Table 2 should be $\mathcal{O}_1 + \mathcal{O}_2 + \mathcal{O}_3$ which is in fact what we find by analyzing the Hilbert series associated to the corresponding orbifold geometry in the following way. Evaluating (59) for the group $G(3,3,2)$ defined by (95) gives

$$
\begin{aligned}
H_{\mathcal{M}_\Gamma}(t,v,\boldsymbol{u}) = \frac{1}{6}\Bigg( & \frac{1}{\left(t^2 u_1^2 v^2 - 2tu_1 v + 1\right)\left(\frac{t^2 u_2^2}{v^2} - \frac{2tu_2}{v} + 1\right)\left(\frac{t^2 u_2^2 v^2}{u_1^2} - \frac{2tu_2 v}{u_1} + 1\right)} \\
& + \frac{2}{\left(t^2 u_1^2 v^2 + tu_1 v + 1\right)\left(\frac{t^2 u_2^2}{v^2} + \frac{tu_2}{v} + 1\right)\left(\frac{t^2 u_2^2 v^2}{u_1^2} + \frac{tu_2 v}{u_1} + 1\right)} \\
& + \frac{3}{\left(1 - t^2 u_1^2 v^2\right)\left(1 - \frac{t^2 u_2^2}{v^2}\right)\left(1 - \frac{t^2 u_2^2 v^2}{u_1^2}\right)} \Bigg),
\end{aligned}
\tag{67}
$$

from which we extract the PLog (64) at first non-trivial order

$$
\mathrm{PLog}_{\mathcal{M}_\Gamma/\mathcal{H}_\Gamma/\mathcal{C}_\Gamma}(t,v,\boldsymbol{u}) = \left(\frac{u_2^2 v^2}{u_1^2} + u_1^2 v^2 + \frac{u_2^2}{u_1} + u_1 u_2 + \frac{u_2^2}{v^2} + u_2 v^2\right)t^2 + O(t^3).
\tag{68}
$$

The coefficient of $t^2$ is seen to be $\mathcal{O}_1 + \mathcal{O}_2 + \mathcal{O}_3$ using the characters (66).

**Example** For another example, look now at #21. Even though the group is much bigger than in the previous example, giving a very prohibitively cumbersome expression for $H_{\mathcal{M}_\Gamma}(t,v,\boldsymbol{u})$, the PLog expansion simplifies as

$$
\mathrm{PLog}_{\mathcal{M}_\Gamma/\mathcal{H}_\Gamma/\mathcal{C}_\Gamma}(t,v,\boldsymbol{u}) = \left(\frac{u_2^2}{u_1} + u_1 u_2\right)t^2 + O(t^3),
\tag{69}
$$

which we identify as $\mathcal{O}_3$. The fact that $c_1 = c_2 = 0$ means that there is no extra supersymmetry, so the corresponding theory is genuinely $\mathcal{N} = 3$. Having $c_3 = 1$ and $c_4 = 0$ shows that this is not a product theory and there is no free sector. Indeed this geometry corresponds to a known $\mathcal{N} = 3$ S-fold.

## 4.1 Constraints and explanation of color shading

Now that we have understood how to interpret the $\mathrm{PLog}_{\mathcal{M}_\Gamma}$, we can state how the constraints on the geometries, reflected by the color coding in the various tables, come about.

a. We first establish whether a given geometry can be obtained as a discrete gauging of a known $\mathcal{N} = 4$ theory. Possible discrete gaugings of interacting field theories are strongly constrained [26] and can be easily listed; see Table 8 below. The corresponding geometries in our tables are shaded in **blue**. All the geometries which do not appear in Table 8 cannot be interpreted this way. Another possibility is that a given geometry could be interpreted as a discrete gauging of a free $U(1) \times U(1)$ $\mathcal{N} = 4$ theory. The analysis of this case is a bit trickier and will be described in the next section, but in short we can establish whether this is the case or not by direct inspection of the irreducible action $\mu_\tau(\Gamma)$. The geometries which can be interpreted as a discrete gauging of a $U(1) \times U(1)$ $\mathcal{N} = 4$ theory are instead shaded in **orange** in the various tables.

b. Next we look at the coefficient of $\mathcal{O}_3$. If $c_3 \geq 2$ then the theory has to be a product theory which implies that the moduli space should be the cartesian product of two rank-1 geometries, $\mathcal{M}_\Gamma = \mathcal{M}_\Gamma^{(1)} \times \mathcal{M}_\Gamma^{(2)}$. Since all such rank-1 geometries have an $\mathcal{N} = 2$ CB with a freely generated coordinate ring, all the entries in Tables 3 or 4 with $c_3 \neq 1$ should be deemed unphysical and are shaded in **red** in the tables. This is a bit too quick though, for it is known that discrete gaugings of interacting theories can give rise to non-freely generated $\mathcal{N} = 2$ CBs [26]. This is also the case for the discrete gauging of $U(1) \times U(1)$ $\mathcal{N} = 4$ (see below) which is a product theory. If a given entry can be interpreted as discrete gauging of the free $U(1)^2$ theory we keep it in the list of consistent geometries and will shade it in **orange**.

c. Finally we look at the coefficient of $\mathcal{O}_4$. The putative theory $\mathcal{T}_\mathcal{M}$ realizing those geometries for which $c_4 \neq 1$ should have a free sector which should give rise again to a factor $\mathcal{M}_{\mathrm{free}} \subset \mathcal{M}_\Gamma$ which factorizes from the rest of the moduli space. If $\dim_\mathbb{C}\mathcal{M}_{\mathrm{free}} = 6$ then $\mathcal{T}_\mathcal{M}$ is a free theory and $\mathcal{M}_\Gamma \equiv \mathcal{M}_{\mathrm{free}} \cong \mathbb{C}^6$. If $\dim_\mathbb{C}\mathcal{M}_{\mathrm{free}} = 3$ then the other factor in the cartesian product should be a rank-1 theory and thus possess a freely generated $\mathcal{N} = 2$ CB slice. The entries in Tables 3 or 4 which cannot be interpreted as a discrete gauging of the $U(1)^2$ $\mathcal{N} = 4$ with $c_4 \neq 0$ are thus unphysical and are also shaded in **red**.

d. Those geometries that pass all the tests described above are listed in Table 1. In Tables 1-4 we shade in **green** those entries which are consistent, cannot be interpreted as discrete gauging of known $\mathcal{N} = 4$ theories and for which no $\mathcal{T}_\mathcal{M}$ is known.

## 5 Known, new and discretely gauged theories

The previous sections list the constraints that an orbifold geometry has to satisfy in order to have a consistent interpretation as the moduli space of a putative $\mathcal{N} = 3$ theory $\mathcal{T}_\mathcal{M}$. In Tables 2 through 4 we have reported all the (principally polarized) rank-2 TSK orbifold geometries: there are 53 of them. We shaded in red the 22 geometries which don't satisfy the extra constraints coming from the study of the $\mathrm{PLog}_{\mathcal{M}_\Gamma}$. A sanity check on the correctness of our analysis is that all moduli spaces of known rank-2 theories should appear in the remaining list of 31 geometries. We will perform this analysis first and find that they in fact all appear. Known theories only realize 23 entries in our tables.

We will then discuss how to interpret the remaining 8 geometries. We show that 2 of these geometries can be interpreted as a straightforward higher-rank generalization of the discrete

gauging in [5,11], but that the remaining 6 geometries cannot be given such an interpretation. We conjecture that they are the moduli spaces of new rank-2 $\mathcal{N} = 3$ conformal field theories. Three of the six have freely-generated Coulomb branch chiral rings, so their $c = a$ central charges can be predicted following [42]. The other three have the remarkable property that they have non-freely generated Coulomb branch chiral rings and they do not arise as discrete gaugings of any known theory. Their $c = a$ central charges are unknown.

For clarity, the results of the analysis of this section are gathered in Tables 6 through 11. All the 31 geometries that are not shaded in red appear at least once in these tables (and sometimes more than once).

It is important to stress once more that we do not make the assumption that a given geometry is realized by a single theory $\mathcal{T}_\mathcal{M}$. In fact it is well-known that this is not the case and geometries can correspond to multiple distinct theories.

## 5.1 Product of rank-1 theories

Let's start by listing all the orbifolds which correspond to the moduli spaces of the product of two rank-1 theories. Recall that in rank 1 there are only 8 admissible scale-invariant geometries which are listed in Table 5. As discussed in detail in [7], entries 5 through 7 do satisfy all requirements to be interpreted as CB slice of a $\mathcal{N} = 3$ moduli space but these spaces fail to be orbifolds as the identification on the flat coordinates by a group element fails to lift to a group action on a smooth space. There are no known $\mathcal{N} = 3$ theories which realize these geometries and, as explained above, the resulting non-orbifold TSK spaces have an unusual $\mathcal{N} = 3$ field content and might be unphysical [7–9].

Table 5: The list of allowed scale-invariant CB geometries for rank-1 $\mathcal{N} \geq 2$ theories. Only the orbifold geometries can be interpreted as $\mathcal{N} \geq 3$ theories. The "Kodaira" column give the Kodaira type of the singularity of the associated Seiberg-Witten curve. In this table we use the term S-fold to really mean *flux-full* S-fold. See the discussion in the "S-folds" paragraph below.

| | Kodaira | Orbifold | $\mathrm{PLog}_\Gamma(t)$ | Corresponding $\mathcal{N} \geq 3$ theory |
|---|---|---|---|---|
| 1. | $II^*$ | $\mathbb{C}/\mathbb{Z}_6$ | $t^6$ | $\mathbb{Z}_3$ gauging of SU(2) $\mathcal{N} = 4$ |
| | | | | $\mathbb{Z}_6$ gauging of $U(1)$ $\mathcal{N} = 4$ |
| 2. | $III^*$ | $\mathbb{C}/\mathbb{Z}_4$ | $t^4$ | $\mathbb{Z}_4$ $\mathcal{N} = 3$ S-fold |
| | | | | $\mathbb{Z}_2$ gauging of SU(2) $\mathcal{N} = 4$ |
| | | | | $\mathbb{Z}_4$ gauging of $U(1)$ $\mathcal{N} = 4$ |
| 3. | $IV^*$ | $\mathbb{C}/\mathbb{Z}_3$ | $t^3$ | $\mathbb{Z}_3$ $\mathcal{N} = 3$ S-fold |
| | | | | $\mathbb{Z}_3$ gauging of $U(1)$ $\mathcal{N} = 4$ |
| 4. | $I_0^*$ | $\mathbb{C}/\mathbb{Z}_2$ | $t^2$ | SU(2) $\mathcal{N} = 4$ |
| | | | | $\mathbb{Z}_2$ gauging of $U(1)$ $\mathcal{N} = 4$ |
| 5. | $IV$ | *Not an orbifold* | | |
| 6. | $III$ | *Not an orbifold* | | |
| 7. | $II$ | *Not an orbifold* | | |
| 8. | $I_0$ | $\mathbb{C}$ | $t$ | $U(1)$ $\mathcal{N} = 4$ |

Each one of the remaining entries in Table 5 is realized as a CB of a known $\mathcal{N} \geq 3$ theory. In fact, each corresponds to multiple theories as reported in the last column of Table 5. Since we are here only interested in listing geometries with known realizations we won't keep track of this extra refinement.

Of course none of these geometries appear in our list directly but many entries are instead realized as the product of two rank-1 theories $\mathcal{T}_1 \times \mathcal{T}_2$. The geometry of the moduli space of

the product theory is the cartesian product $\mathcal{M}_\Gamma = \mathcal{M}_{\Gamma_1} \times \mathcal{M}_{\Gamma_2}$ of the moduli spaces of the individual spaces. Knowing the PLog of the individual geometries we can straightforwardly compute the PLog of product theories $\mathcal{T}_1 \times \mathcal{T}_2$ as $\mathrm{PLog}(\mathcal{T}_1 \times \mathcal{T}_2) = \mathrm{PLog}(\mathcal{T}_1) + \mathrm{PLog}(\mathcal{T}_2)$. As mentioned above various times, all the rank-1 geometries have a freely generated coordinate ring and so do the rank-2 geometries constructed as cartesian product of them. The entries of Table 2 which are realized as product of rank-1 theories are reported in Table 6.

Table 6: The list of entries in Table 2 which are interpreted as products of two rank-1 theories.

| Entry # | | Geometry | Entry # | | Geometry |
|---|---|---|---|---|---|
| 1 | $\longrightarrow$ | $I_0 \times I_0$ | 11 | $\longrightarrow$ | $IV^* \times IV^*$ |
| 2 | $\longrightarrow$ | $I_0 \times I_0^*$ | 13 | $\longrightarrow$ | $I_0^* \times II^*$ |
| 3 | $\longrightarrow$ | $I_0 \times IV^*$ | 14 | $\longrightarrow$ | $IV^* \times III^*$ |
| 4 | $\longrightarrow$ | $I_0^* \times I_0^*$ | 15 | $\longrightarrow$ | $III^* \times III^*$ |
| 5 | $\longrightarrow$ | $I_0 \times III^*$ | 17 | $\longrightarrow$ | $IV^* \times II^*$ |
| 7 | $\longrightarrow$ | $I_0^* \times IV^*$ | 20 | $\longrightarrow$ | $IV^* \times II^*$ |
| 8 | $\longrightarrow$ | $I_0 \times II^*$ | 22 | $\longrightarrow$ | $II^* \times II^*$ |
| 10 | $\longrightarrow$ | $I_0^* \times III^*$ | | | |

## 5.2 Known genuinely rank-2 theories

Now consider the genuinely rank-2 theories whose moduli space does not factorize as the product of two rank-1 geometries. We analyze separately $\mathcal{N} = 4$ theories, $\mathcal{N} = 3$ theories which are realized as S-folds, and $\mathcal{N} = 3$ theories which are realized as discrete gaugings of $\mathcal{N} = 4$ theories.

$\mathcal{N} = 4$ **theories.** As discussed in section 2, the distinguishing feature of $\mathcal{N} = 4$ theories is the existence of a dimension two generator of the CB coordinate ring. This translates into the presence of a $t^2$ term in the $\mathrm{PLog}_\Gamma(t)$ of the corresponding CB geometry.[16] Scrolling down the various tables, one finds that there is a one to one correspondence between the number of $t^2$ terms in $\mathrm{PLog}_\Gamma(t)$ of a given geometry and the complex dimensionality of the fixed point locus in $\mathfrak{H}_2$. This reflects the fact outlined above that for each generator of dimension 2 in the CB chiral ring there exists an associated exactly marginal operator.

Since the moduli space geometry of an $\mathcal{N} = 4$ gauge theory only depends on the gauge Lie algebra and neither on the global form of the gauge group nor on the spectrum of line operators, we expect only three non-product rank-2 $\mathcal{N} = 4$ theories corresponding to the lagrangian theories with gauge Lie algebras $\mathfrak{su}(3)$, $\mathfrak{so}(5) \cong \mathfrak{sp}(2)$ and $G_2$. The corresponding geometries are listed in the first part of Table 9.

As discussed in [14], $\mathcal{N} = 4$-preserving discrete gauging of interacting $\mathcal{N} = 4$ gauge theories at rank 2 does not produce any other inequivalent CB geometry, so the above 3 geometries might be expected to exhaust the list of $\mathcal{N} = 4$ genuinely rank-2 geometries. But Tables 3 and 4 show many more geometries with a $t^2$ term in their $\mathrm{PLog}_\Gamma(t)$. These geometries do correspond in fact to $\mathcal{N} = 4$ theories but arise via a non-trivial generalization of $\mathcal{N} = 4$-preserving discrete gauging which only applies to product of Maxwell (i.e. $U(1)$ with no charged matter) theories. To our knowledge this generalization has not appeared elsewhere and thus deserves a separate discussion. For this reason we will discuss these geometries in section 5.3 below.

---

[16]As explained in the previous section, the supersymmetry enhancement can be also seen from the study of the $\mathrm{PLog}_{\mathcal{M}_\Gamma}$ as a non-zero $c_1 = c_2$. This condition is completely analogous to the one discussed in the text.

So far we have accounted for the first 15 entries in Table 2 and entries 17, 20 and 22. Let's now turn to discuss those geometries which can be interpreted as a moduli space of rank-2 $\mathcal{N} = 3$ theories.

**S-folds.** The first class of $\mathcal{N} = 3$ theories to have been constructed where engineered in F-theory using the *S-fold* construction [11, 12]. An S-fold is roughly a generalization of an orientifold which acts on its transverse space as $(\mathbb{C}^3 \times T^2)/\mathbb{Z}_m$, for $m = 1, 2, 3, 4, 6$.[17] They come in two variants depending on whether a discrete torsional flux is turned on. The worldvolume theory of $r$ D3 branes probing an S-fold is a rank $r$ four dimensional SCFT with $\mathcal{N} \geq 3$ supersymmetry. Their $\mathcal{N} = 2$ CBs are the orbifolds $\mathbb{C}^r/G(m, 1, r)$ or $\mathbb{C}^r/G(m, m, r)$ for the *flux-full* and *flux-less* S-fold respectively. It was argued in [12] that not all $m$ admit the flux-full variant and for $m = 6$ only the flux-less one is allowed (which is the reason why there is no S-fold in the entry 1 in table 5). For $r = 1$ the theories obtained by probing flux-less S-folds can be engineered as discretely gauged version of $U(1)$ $\mathcal{N} = 4$ Lagrangian theories, while for $r = 2$ they give rise to genuine rank-2 $\mathcal{N} = 4$ theories. We here use this fact, i.e. that flux-less S-folds can be understood as variant of Lagrangian theories for $r = 1, 2$, to avoid altogether to refer to the flux.[18] We will then use the term S-folds to solely refer to the flux-full case.[18] The list of CB orbifold geometries for S-folds is given in Table 7.[19]

Table 7: List of geometries corresponding to rank-2 S-folds. The first four lines correspond to $\mathcal{N} = 4$ theories (this phenomenon is specific to rank-2 S-folds) and already appear in Table 9. Only the last two lines correspond to $\mathcal{N} = 3$ S-folds.

| Entry # | | S-fold orbifold group |
|:---:|:---:|:---:|
| 4 | $\longrightarrow$ | $G(2, 2, 2)$ |
| 6 | $\longrightarrow$ | $G(3, 3, 2)$ |
| 9 | $\longrightarrow$ | $G(4, 4, 2) = G(2, 1, 2)$ |
| 12 | $\longrightarrow$ | $G(6, 6, 2)$ |
| 18 | $\longrightarrow$ | $G(3, 1, 2)$ |
| 21 | $\longrightarrow$ | $G(4, 1, 2)$ |

**Discrete gaugings of interacting $\mathcal{N} = 4$ theories.** Discrete gaugings of $\mathcal{N} = 4$ Yang-Mills theories which preserve $\mathcal{N} = 3$ supersymmetry [13, 14] correspond to gauging certain $\mathbb{Z}_k$ global symmetries with $k = 2, 3, 4, 6$. In those cases, the orbifold group is $W(\mathfrak{g}) \rtimes \mathbb{Z}_k$, where $W(\mathfrak{g})$ is the Weyl group of the $\mathcal{N} = 4$ gauge algebra $\mathfrak{g}$. They correspond to entries 16, 18, 23 and 42 of our tables. Note however that not all values of $k$ are allowed for any Lie algebra $\mathfrak{g}$. In fact the gaugeable subgroups depend on the detailed form of the S-duality group of the various theories [47, 48]. For instance the S-duality group of the $\mathfrak{su}(3)$ $\mathcal{N} = 4$ theory only has appropriate $\mathbb{Z}_k$ symmetries with $k = 2, 3, 6$. This is in agreement with our list of solutions, which do not have an entry which would correspond to $W(\mathfrak{su}(3)) \rtimes \mathbb{Z}_4$, whose $\text{PLog}_\Gamma(t)$ would be equal to $t^4 + t^8 + t^{12} - t^{16}$.

---

[17]Strictly speaking, the term S-fold is reserved to $m = 3, 4, 6$; the case $m = 1$ corresponds to no identification at all, and $m = 2$ is an actual orientifold plane.

[18]Flux-less S-folds can give rise to genuinely new theories even at rank 2; the worldvolume theory on D3 branes probing the S-folds combined with exceptional sevenbranes gives rise to $\mathcal{N} = 2$ SCFTs [43]. In this case the flux-less S-folds give genuinely new $\mathcal{N} = 2$ SCFTs even in the presence of only two D3 probes [44–46].

[19]In [12] it was noted that rank-2 $\mathcal{N} = 4$ theories can also be realized as S-folds. This is why we include these theories in Table 7.

Table 8: List of geometries corresponding to rank-2 $\mathcal{N} = 3$ theories obtained from discrete gaugings of $\mathcal{N} = 4$ Yang-Mills theories.

| Entry # | | Theory |
|---|---|---|
| 16 | $\longrightarrow$ | $\mathbb{Z}_2$ gauging of $\mathfrak{sp}(2)$ $\mathcal{N} = 4$ |
| 18 | $\longrightarrow$ | $\mathbb{Z}_3$ gauging of $\mathfrak{su}(3)$ $\mathcal{N} = 4$ |
| 23 | $\longrightarrow$ | $\mathbb{Z}_6$ gauging of $\mathfrak{su}(3)$ $\mathcal{N} = 4$ |
| | | $\mathbb{Z}_3$ gauging of $G_2$ $\mathcal{N} = 4$ |
| 42 | $\longrightarrow$ | $\mathbb{Z}_3$ gauging of $\mathfrak{sp}(2)$ $\mathcal{N} = 4$ |

### 5.3 New rank-2 theories

Tables 6-8 list 23 of the 31 consistent geometries in Tables 2-4 and (at least one) corresponding theory $\mathcal{T}_\mathcal{M}$ realizing them. This section is dedicated to the interpretation of the remaining 8 geometries which do not correspond to any theory previously constructed. 2 of them can be interpreted as discrete gaugings of the $U(1)^2$ $\mathcal{N} = 4$ Maxwell theory and are thus moduli spaces of free theories despite their complicated algebraic structure. 3 of the remaining 6 have an $\mathcal{N} = 2$ CB with a freely-generated coordinate ring but do not correspond to any known theory. We speculate that a generalization of the S-fold construction might realize them. Finally three geometries have instead a non-freely-generated $\mathcal{N} = 2$ CB and thus their conjectural associated $\mathcal{N} = 3$ conformal field theories will belong to a novel class. They would provide the first example of theories with a non-freely-generated $\mathcal{N} = 2$ CB chiral ring but having a trivial 2−form symmetry (that is, they cannot be realized by gauging a 0-form symmetry). We will now discuss each of these three types of new geometry in detail.

**Rank-2 discrete gauging of** $U(1)^2$ **Maxwell theory.** We start with the most boring possibility: geometries which can be interpreted as a moduli space of a discretely gauged $U(1)^2$ $\mathcal{N} = 4$ free Maxwell theory. We will denote them as $[\,U(1)^2\,]_{\tilde{\Gamma}}$, where $\tilde{\Gamma}$ is the finite subgroup of $SO(6)_R \times Sp(4, \mathbb{Z})$ which we gauge. All of these theories are free and thus not of much physical interest. However, the analysis here will be useful for our purposes since it will enable us to identify which of the geometries *cannot* be interpreted this way.

There are a few subtleties in generalizing the discussion of discrete gauging [5,11,13,14] to this free rank-2 example. The R-symmetry group of a $U(1)^2$ $\mathcal{N} = 4$ theory is $SO(6)_R^{\text{diag}}$, the diagonal subgroup of $SO(6)_R^1 \times SO(6)_R^2$, where $SO(6)_R^i$ acts on the supercharges implementing the supersymmetry transformation of the $i$-th $U(1)$. In addition, there is an $Sp(4, \mathbb{Z})$ UV EM duality group acting on the three complex dimensional conformal manifold of the theory parameterized by $\tau_{ij}$.[20] The theory is free and has no charged states in the spectrum, thus the only observables which can distinguish theories with different values of $\tau_{ij}$ are their response to infinitely massive charged probes. We must include them if we are to be able to analyze how the theory transforms under the action of the S-duality group.

Call $\tau_{ij}^{\text{UV}}$ the value of the holomoprhic gauge coupling in the UV. If all charged states are infinitely massive, then there is no RG-running and $\tau_{ij}^{\text{UV}} = \tau_{ij}^{\text{IR}}$. Furthermore no degree of freedom decouples in the running and thus the effective Lagrangian at very low energy is in fact the UV Lagrangian. It is known that an action of the S-duality group induces an action of the low-energy EM-duality group, see for instance [49]. In this case the two groups simply coincide, and we will denote by $M \in Sp(4, \mathbb{Z})$ a generic element of these identical groups.

---

[20]If $\tau_{12} = 0$, the two theories are completely decoupled and it is possible to talk about two separate $\mathcal{N} = 4$ algebras. In this picture the R-symmetry is enhanced to the full $SO(6)_R^1 \times SO(6)_R^2$ but the EM duality group is $SL(2, \mathbb{Z})^1 \times SL(2, \mathbb{Z})^2 \subset Sp(4, \mathbb{Z})$. This is the right picture to describe discrete gauging giving rise to product theories as we can act on the two separate $U(1)$s independently.

As argued in [49], the action of the S-duality group also induces a non trivial action on the supercharges which can always be chosen to commute with the $SU(4)_R$ action. Thus the S-duality group acts as a phase common to all of the supercharges.[21] This means that there exists a morphism $\exp(-i\widehat{\phi}) \colon Sp(4,\mathbb{Z}) \to U(1)$ such that under $M \in Sp(4,\mathbb{Z})$,

$$Q_I \to \exp(-i\widehat{\phi}(M))Q_I \tag{70}$$

for all $I$. We can use the identification of the S-duality with EM duality action to explicitly compute it.

The bosonic part of the Lagrangian of the $\mathcal{N} = 4$ $U(1)^2$ theory is

$$\mathcal{L}_{\text{bosonic}} = \text{Im}\left[\tau^{ij}(a)\left(\partial a_i^{IJ} \cdot \partial \overline{a}_{IJj} + \mathcal{F}_i \cdot \mathcal{F}_j\right)\right], \qquad I,J = 1,...,4, \quad i,j = 1,2. \tag{71}$$

Here we use complex variables satisfying the reality condition $\epsilon^{IJKL}\overline{a}_{KLj} = a_j^{IJ}$. This is completely analogous to (6) with the only modification that the capital indices are $SU(4)_R$ and not $U(3)_R$ indices. Again here the $\mathcal{F}_i$ are the self-dual $U(1)$ which are related to the $a_i^{IJ}$ (and not to the $\overline{a}_{IJj}$) by supersymmetry as [7],

$$\epsilon^{IJKL}\mathcal{F}_i \sim Q^I Q^J a_i^{KL}. \tag{72}$$

$M$ induces a transformation on $\tau$ via (35), on the $a_i^{IJ}$ via $\mu_\tau(M)$ (36), and on the supercharges via $\exp(-i\widehat{\phi}(M))$. From (72) we infer that the $\mathcal{F}_i$ transform as

$$\mathcal{F}_i \to \exp(-2i\widehat{\phi}(M))\mu_\tau(M)_i{}^j \mathcal{F}_j. \tag{73}$$

The phase $\widehat{\phi}(M)$ is defined up to an action of the $\mathbb{Z}_4$ center of $SU(4)_R$ and thus henceforth we consider $\widehat{\phi}(M) \in [0, \pi/2)$. Note that the map $\mu_\tau$ is a group homomorphism only for the subgroup of $Sp(4,\mathbb{Z})$ which fixes $\tau$.

We will now identify a set of necessary conditions which allow us to identify those geometries which cannot be interpreted as $[U(1)^2]_{\tilde{\Gamma}}$.

S-duality is an equivalence between different descriptions of the same theory: different holomorphic gauge couplings describe the same physics. Thus it is not a global symmetry which maps distinct operators of a given description of the theory, and so gauging subgroups of the S-duality group does not make sense in general. The situation is different for those finite subgroups $\Gamma \subset Sp(4,\mathbb{Z})$ which fix some value of the coupling, $\tau_{\text{fix}}$, and thus act within a single description of the theory. In the case of non-product theories and with the coupling set to $\tau = \tau_{\text{fix}}$, such $\Gamma$ may act as global symmetries which could then be gauged.

However, it turns out that such a $\Gamma \subset Sp(4,\mathbb{Z})$ fixing the $\tau$ of an $\mathcal{N} = 4$ $U(1)^2$ free Maxwell theory can fail to be a global symmetry of the theory. We can check this by computing the action of $\Gamma$ on the lagrangian (71). The first term in (71) is always preserved by the $\Gamma$ action.[22] But demanding the invariance of the second term gives, using (73), the non-trivial condition

$$\exp(-4i\widehat{\phi}(M))\,\mu_\tau(M)^T\,\tau\,\mu_\tau(M) = \tau. \tag{74}$$

A solution exists only if $\mu_\tau(M)^T\,\tau\,\mu_\tau(M)\,\tau^{-1}$ is proportional to $\mathbb{1}_2$ for all $M \in \Gamma$.

As discussed extensively in sections 2 and 3, the orbifold geometries in Tables 2-4 are precisely in one-to-one correspondence with subgroups $\Gamma \subset Sp(4,\mathbb{Z})$ and their corresponding $\tau_{\text{fix}}$. Only a subset of all the $\Gamma$ groups admit a solution for (74).

---

[21]The $U(1)$ which acts as a common phase multiplication on the $Q^I$s was called *chiral rotation* in [49].

[22]While the calculation is slightly non-trivial, the result should be expected: the first term in (71) gives the metric on the orbifold geometry, $\mu_\tau(\Gamma)$ gives the orbifold action, and the orbifold construction only works because $\mu_\tau(\Gamma)$ is in fact an isometry.

But even if a solution does exist for a given $\Gamma$ and its fixed $\tau$, and so it acts as a global symmetry of the $\mathcal{N} = 4$ Maxwell theory which can be gauged, this is not the end of the story. For (74) also then determines the phase, $\exp(-4i\widehat{\phi}(M))$, by which the supercharges transform. If we were to gauge $\Gamma$, because of the non-trivial phase $\widehat{\phi}$, we would break supersymmetry completely. In order to preserve $\mathcal{N} = 3$ supersymmetry we must gauge instead the combination the $\Gamma$ action with that of the $\mathrm{SO}(6)_R$ elements

$$\mathcal{R}(M) = \begin{pmatrix} R_{2\widehat{\phi}(M)} & & \\ & R_{2\widehat{\phi}(M)} & \\ & & R_{-2\widehat{\phi}(M)} \end{pmatrix}, \qquad (75)$$

where $R_{2\widehat{\phi}(M)}$ implements an $\mathrm{SO}(2)$ rotation by $2\widehat{\phi}(M)$. These R-symmetry rotations then induce phase transformations of the supercharges which cancel the $\exp(-4i\widehat{\phi}(M))$ phase for at least three of the four supercharges.

Since $\mathcal{R}(M)$ acts non-trivially on the $a_i^{IJ}$s by phase multiplication of the three complex scalar combinations, it acts non-trivially on the moduli space. Thus the gauging of these $\mathrm{SO}(6)_R$ transformations further modifies the resulting moduli space of the discretely gauged theory.

Putting this all together, the correct action of the $\mathcal{N} = 3$-preserving discrete symmetry on the moduli space is

$$\chi : \Gamma \to \mathrm{GL}(2, \mathbb{C}), \qquad \chi(M) := \exp(2i\widehat{\phi}(M))\mu_\tau(M). \qquad (76)$$

Thus, upon gauging $\Gamma$, we end up with the following moduli space geometry (here we are only focusing on an $\mathcal{N} = 2$ CB slice of the moduli space)

$$\mathcal{C} = \mathbb{C}^2/\chi(\Gamma). \qquad (77)$$

By explicit computation we can go through the list of $\Gamma$, identify those for which a solution of (74) exists and then compute $\mathbb{C}^2/\chi(\Gamma)$. Those entries in Tables 2-4 which do not have any other known construction but whose CB coincides with one of the $\mathbb{C}^2/\chi(\Gamma)$ thus computed are shaded in **orange**. They are entries 26 and 28 in Table 9. These have either $c_3 \geq 2$ or $c_4 \neq 0$, as might be expected of a discrete gauging of a free theory.

Table 9: List of geometries corresponding to rank 2 $\mathcal{N} = 4$ theories (excluding product theories).

| Entry # | | Theory |
|---------|---|--------|
| 6 | $\longrightarrow$ | $\mathcal{N} = 4$ with $\mathfrak{g} = \mathfrak{su}(3)$ |
| 9 | $\longrightarrow$ | $\mathcal{N} = 4$ with $\mathfrak{g} = \mathfrak{so}(5) \cong \mathfrak{sp}(2)$ |
| 12 | $\longrightarrow$ | $\mathcal{N} = 4$ with $\mathfrak{g} = G_2$ |
| 26 | $\longrightarrow$ | $\mathbb{Z}_2$ gauging of $\mathcal{N} = 4$ with $\mathfrak{g} = \mathfrak{u}(1) \oplus \mathfrak{u}(1)$ |
| 28 | $\longrightarrow$ | $\mathbb{Z}_4$ gauging of $\mathcal{N} = 4$ with $\mathfrak{g} = \mathfrak{u}(1) \oplus \mathfrak{u}(1)$ |

While the set of conditions outlined above seem a reasonable set of necessary conditions to identify the set of geometries which can be interpreted as arising from discrete gauging of the free $\mathrm{U}(1)^2$ $\mathcal{N} = 4$ theory, some of the $\mathbb{C}^2/\chi(\Gamma)$ orbifolds we find from the (76) action are surprising. In particular, we expected that all the $\mathbb{C}^2/\chi(\Gamma)$ should appear in our list of possible $\mathcal{N} = 3$ orbifold TSK geometries since the discrete gauging procedure we have outlined preserves $\mathcal{N} = 3$ supersymmetry and all the low energy conditions which led to Tables 2-4. But instead we find (by direct computation) that some of the $\mathbb{C}^2/\chi(\Gamma)$ orbifolds do not appear in this set, because the map (76) spoils the crystallographic condition of the initial $\Gamma$ action. We don't understand how to interpret this phenomenon and we leave this question for future studies.

$\mathcal{N} = 3$ **theories from complex reflection groups.** TSK orbifold geometries with freely generated coordinate ring are in one-to-one correspondence with with *crystallographic complex reflection groups* (CCRG) [34] which preserve a principal polarization [7, 10]. These groups are exactly the 25 groups listed in Table 2. Excluding the geometries corresponding to products of rank-1 theories, $\mathcal{N} = 4$ theories and its discretely gauged versions, and S-folds, there are three new geometries, listed in Table 10. Let's discuss briefly the interpretation of these geometries.

Since the $\mathcal{N} = 2$ CB is freely generated and since by $\mathcal{N} = 3$ supersymmetry $a = c$, the Shapere-Tachikawa method of computing central charges [42] applies to this case unambiguously, giving

$$a = c = \sum_{i=1}^{2} \frac{2\Delta_i - 1}{4}, \tag{78}$$

where $\Delta_1$ and $\Delta_2$ are the scaling dimension of the generators of the CB chiral ring which can be read off from the exponents of the $\mathrm{PLog}_\Gamma(t)$ polynomial of the corresponding entries. Applying 78 we obtain the values reported both in Tables 1 and 10.

Table 10: List of geometries corresponding to CCRGs and which are not products of rank-1 theories nor $\mathcal{N} = 4$ theories nor S-folds, and their corresponding central charges. $\mathrm{ST}_{12}$ is isomorphic to the binary octahedral group.

| Entry # | | $\Gamma$ | $4c = 4a$ |
|---------|--|----------|-----------|
| 19 | $\longrightarrow$ | $G(6, 3, 2)$ | 18 |
| 24 | $\longrightarrow$ | $\mathrm{ST}_{12}$ | 26 |
| 25 | $\longrightarrow$ | $G(6, 1, 2)$ | 34 |

By a more detailed analysis of these theories, analogous to what we carried out in section 2.3, we could extract more information about the physics. In particular we could compute the number of components of the singular locus, the monodromies around them and thus gain a partial understanding on the BPS spectrum of these theories. But we won't do it here.

While it is conceivable that a generalization of the S-fold construction of [11, 12] might realize entries # 19 and 25, it is hard to see how such a construction could give a theory corresponding to geometry # 24. This geometry is the only rank 2 geometry obtained by one of the exceptional CCRG groups,[23] and thus there is no higher rank version of such a theory, as would be obtained if it were realized by probing an S-fold-like singularity with a arbitrary numbers of D3 branes. This argument, however, should not be taken too literally as it is already known that there are special phenomena, like supersymmetry enhancement in S-folds, that only happen at a particular rank. A more detailed analysis of this theory will appear elsewhere [50].

**New theories.** We will now discuss the last but possibly most interesting geometries we find: those which can consistently be interpreted as interacting rank 2 $\mathcal{N} = 3$ field theories but whose $\mathcal{N} = 2$ CB chiral ring is not freely generated and whose CB slice is a hypersurface in $\mathbb{C}^3$ with a (complex) singularity at the origin. The three geometries are reported in Table 11 and the explicit algebraic form for the CB can be straightforwardly obtained from the expression of the $\mathrm{PLog}_\Gamma(t)$ in Table 3 and will be discussed shortly. Since the finite groups which arise in these geometries do not have standard accepted names, it is useful to explicitly give a presentation of these groups and some information about the size of their conjugacy classes

---

[23]The classification of CCRGs is in ways analogous to the simple Lie algebra, with few infinite series and some exceptional entries. Moduli spaces corresponding to exceptional CCRGs only exist up to rank 6.

and orders of their elements:

$$\text{SD}_{16} \quad : \quad \begin{cases} \langle a, b | a^8 = b^2 = 1, bab = a^3 \rangle \\ \text{order} : \{1, 2^5, 4^6, 8^4\} \\ \text{conj. classes } \{1^2, 2^3, 4^2\} \end{cases} \tag{79}$$

$$M_4(2) \quad : \quad \begin{cases} \langle a, b | a^8 = b^2 = 1, bab = a^5 \rangle \\ \text{order} : \{1, 2^3, 4^4, 8^8\} \\ \text{conj. classes } \{1^4, 2^6\} \end{cases} \tag{80}$$

$$\text{Dic}_3 \times \mathbb{Z}_3 \quad : \quad \begin{cases} \langle a, b, c | a^3 = b^6 = 1, c^2 = b^3, ab = ba, cac^{-1} = a^{-1}, cbc^{-1} = b^{-1} \rangle \\ \text{order} : \{1, 2, 3^8, 4^6, 6^8, 12^5\} \\ \text{conj. classes } \{1^6, 2^6, 3^6\} \end{cases} \tag{81}$$

Here the notation $x^y$, used both for the order of the elements and the size of the conjugacy classes, means that the entry $x$ repeats $y$ times in the list.

As we have mentioned many times, the abstract presentations of the groups provided above do not characterize the orbifold geometries. Only by using their actions on $\mathbb{C}^2$ can we compute the algebraic form of the $\mathcal{N} = 2$ CB of each one of these geometries as a hypersurface in $\mathbb{C}^3$. We find

$$\begin{aligned} \mathcal{C}_{\text{SD}_{16}} &:= \left\{ (u_4, u_6, u_8) \in \mathbb{C}^3 | u_4 u_8 + u_6^2 = 0 \right\}, \\ \mathcal{C}_{M_4(2)} &:= \left\{ (u_4, u_8, \tilde{u}_8) \in \mathbb{C}^3 | \tilde{u}_8 u_8 + u_4^4 = 0 \right\}, \\ \mathcal{C}_{\text{Dic}_3 \times \mathbb{Z}_3} &:= \left\{ (u_6, u_{12}, \tilde{u}_{12}) \in \mathbb{C}^3 | u_{12} \tilde{u}_{12} + u_6^4 = 0 \right\}. \end{aligned} \tag{82}$$

Again we could perform an analysis of these geometries along the lines of section 2.3 and learn about their discriminant locus and possibly obtain partial information on their BPS spectrum. It is unfortunately impossible to get any prediction for their central charges. This is related to something that was touched upon in section 3.5, that is the fact that if the coordinate ring of the CB is not freely generated, there isn't a unique, nor even well-defined, notion of primary generators of the ring of invariant polynomials. In other words there isn't a canonical choice of $\Delta_1$ and $\Delta_2$ which could be plugged into the Shapere-Tachikawa formula (78). In more generality, the analysis of [42] assumes that the CB is freely generated and thus does not apply to this case.

Table 11: The three geometries in Table 3 which could be interpreted as moduli spaces of rank-2 interacting $\mathcal{N} = 3$ theories but whose $\mathcal{N} = 2$ CB chiral ring is non-freely generated. No information on the central charges of these theories is available.

| Entry # | | $\Gamma$ |
|---|---|---|
| 38 | $\longrightarrow$ | $\text{SD}_{16}$ |
| 39 | $\longrightarrow$ | $M_4(2)$ |
| 43 | $\longrightarrow$ | $\text{Dic}_3 \times \mathbb{Z}_3$ |

One conceivable way to compute the central charge for these theories, is to study the corresponding VOA (more below) by guessing a set of operators and hope that their algebra closes only for a single value of the central charges, as in the case for the VOA corresponding to rank-1 $\mathcal{N} = 3$ theories [51]. The closing of the VOA would corroborate further the hypothesis of the existence of these truly exotic $\mathcal{N} = 3$ theories corresponding to the geometries in Table 11. In any case the existence and the consistency of these geometries urge further and deeper studies of possible realization of $\mathcal{N} = 3$ theories to either construct theories realizing Table 11 or disprove their existence.

# 6 Conclusion and open questions

In this manuscript we have carried out the analysis of the rank 2 geometries which can be interpreted as moduli spaces of $\mathcal{N} = 3$ theories. A crucial assumption that we make is that all such geometries are orbifolds of $\mathbb{C}^{3r}$ though, as explained in [7], it remains an open question whether this is in fact the case.

Many of the geometries correspond either to known theories or can be interpreted as the moduli space of discretely gauged versions of known theories. And the moduli space geometries of all known rank 2 $\mathcal{N} \geq 3$ theories do appear in our classification.

But, remarkably, we find six geometries which are not realized by any known theory and thus predict the existence of new $\mathcal{N} = 3$ theories, three of which have the exotic property of having a non-freely generated CB chiral ring. If the existence of these theories is confirmed, they would provide the first example of theories with a non-freely generated CB chiral ring not obtained as discretely gauged version of a theory with a freely generated CB ring. This in turn would further strengthen our belief that the set of $\mathcal{N} = 2$ SCFTs with a CB geometry isomorphic to $\mathbb{C}^r$ which corresponds, with the exceptions of extremely few cases, to the entirety of $\mathcal{N} = 2$ SCFTs discussed in the literature, is only a subset of the existing theories.

It is of course possible that some or all the new geometries we have constructed here do not correspond to any physical theory. This possibility might arise because the TSK analysis carried out here only captures a subset of physical consistency requirements and thus some of the geometries we label as admissible are in fact unphysical.[24] It would thus be very interesting to extend the present work by further studying the physical requirements on the moduli space of $\mathcal{N} = 3$ theories. We list some possible directions below.

**Higgs branch data and associated VOA.** It is known that every $\mathcal{N} = 2$ conformal theory comes equipped with an intricate structure, the associated vertex operator algebra (VOA) [52–54]. In the following we might also refer to the VOA as chiral algebra. While it is unclear how to characterize VOA of general $\mathcal{N} = 2$ SCFTs, in the case of $\mathcal{N} = 3$ theories things might be considerably more constrained. Firstly the 2d chiral algebra has an extended $\mathcal{N} = 2$ super-Virasoro symmetry, and secondly the ansatz that the generators of the VOA can be fully characterized from Higgs branch data, has worked remarkably well in all known $\mathcal{N} = 3$ examples where the VOA construction has been carried out explicitly. In particular a proposal for how to construct the VOA of $\mathcal{N} = 3$ theories with a freely-generated $\mathcal{N} = 2$ Coulomb branch (those associated to orbifold geometries by complex reflection groups) was outlined in a recent paper [41]. The authors of [41] have come up with a remarkably simple proposal using a free-field realization which only relies on information extracted from the Hilbert series of the Higgs branch.[25] Using (15) and (63), this information can be readily extracted for all geometries considered here. The construction of a consistent chiral algebra with the right central charges and null states would give further evidence for the physical consistency of the new geometries we find.

**Mass deformation.** $\mathcal{N} = 3$ and $\mathcal{N} = 4$ superconformal field theories do not have any relevant deformation, though there exist $\mathcal{N} = 2$-preserving mass deformations. $\mathcal{N} = 4$ theory with those masses turned on are generally referred as $\mathcal{N} = 2^*$ theories. Since the first $\mathcal{N} = 2$ papers by Seiberg and Witten [56], it has been evident that the study of an $\mathcal{N} = 2^*$ theory

---

[24]The fact that purely geometric data is insufficient for physical consistency is discussed at in [7] in the non-orbifold case.

[25]For an application of similar techniques to $\mathcal{N} = 2$ theories and a discussion of the applicability of free-field construction in VOAs see [55].

illuminates the original theory with enhanced supersymmetry. That the study of mass deformations can teach us about the physics of the corresponding conformal theories has been even more the case in the analysis of the rank-1 $\mathcal{N} = 2$ SCFTs carried out in [1,2,4]. This series of papers almost exclusively focuses on the analysis of mass deformations, including those which break $\mathcal{N} = 3$ to $\mathcal{N} = 2$ [3]. Currently we do not know how to generalize the incredibly constraining analysis of mass deformations of rank 1 theories to higher ranks. But it is certainly likely that understanding the behaviour of the geometries in Tables 2-4 after turning on an $\mathcal{N} = 2$-preserving mass deformation might not only give considerable insight into the physics of these theories, but also might further constrain the set of physically consistent geometries.

**Non-principally polarized Dirac pairing.** Our analysis can be fairly straightforwardly extended to theories with non-principal Dirac pairing. This is particularly the case for orbifolds generated by CCRG [10]. Non-principal Dirac pairings are very little discussed in the literature and correspond to theories with a non-standard, and for higher ranks possibly not uniform, normalization of electric and magnetic charges. It would be interesting to clarify the physical properties of such theories. The analysis of rank-1 $\mathcal{N} = 2$ geometries show that allowed normalizations are very constrained, in fact in the rank-1 case there is a single allowed geometry with a non-principal Dirac pairing. This geometry corresponds to a theory with intriguing properties which have not been fully understood yet. A possible interpretation is that the theory with non-principal Dirac pairing is not a genuine field theory but rather a relative one [20].

A naive study of the Dirac pairing induced by 6d (2,0) theories compactified on Riemann surfaces shows that non-principal choices might be allowed. It is well known that 6d (2,0) theories are also relative field theories whose 7 dimensional bulk theory has been constructed explicitly [57]. In order for the 4d theory obtained by compactification of a 6d theory to have a partition function, extra structures need to be specified [57–59]. Perhaps consideration of these subtleties might give insights into theories with non-principal Dirac pairing.

# Acknowledgments

It is a pleasure to thank Jacques Distler, Behzat Ergun, Amihay Hanany, Carlo Meneghelli, Elli Pomoni and Leonardo Rastelli for useful discussions. AB would like to thank Julius Grimminger for precious help in understanding Gottschling's papers. PCA was supported in part by DOE grant DE-SC0011784 and by Simons Foundation Fellowship 506770. The work of AB is supported by STFC grant ST/P000762/1 and grant EP/K034456/1. MM was supported in part by NSF grant PHY-1151392 and in part by NSF grant PHY-1620610.

# A   The Du Val nomenclature

We describe the finite subgroups of $U(2)$ that are used in the text. We adopt the notation introduced by Du Val in [29], and summarized in Table 12. In this notation, the groups are written in the form $(L/L_K; R/R_K)$, where $L \subset U(1)$ and $R \subset SU(2)$ are finite subgroups, and where $L_K$ and $R_K$ are normal subgroups of $L$ and $R$, respectively, such that the quotients $L/L_K$ and $R/R_K$ are isomorphic. We choose an explicit isomorphism $\phi : L/L_K \to R/R_K$. The subgroup of $U(2) = U(1) \times SU(2)$ corresponding to the label $(L/L_K; R/R_K)$ is then

$$\{(l, r) \in L \times R \mid \phi(\bar{l}) = \bar{r}\}, \tag{83}$$

where the bar denotes the projection to the quotient groups. It should be pointed out that the enumeration found in [29] suffers from omissions and repetitions [60]. The complete list of

Table 12: List of the finite subgroups of U(2) used in this paper (note for the list of finite U(2) subgroups to be complete, we would need to include three more families $DV_6(m)$, $DV_7(m)$ and $DV_9(m)$). Here $m$ and $n$ are arbitrary positive integers, except when some restrictions are explicitly noted.

| Name | Group | Explicit form | Order |
|---|---|---|---|
| $DV_1(m,n,r,s)$  (with (85)) | $(\mathbb{Z}_{2mr}/\mathbb{Z}_{2m};\mathbb{Z}_{2nr}/\mathbb{Z}_{2n})_s$ | (84) | $\frac{1}{2}mnr$ |
| $DV_2(m,n)$ | $(\mathbb{Z}_{2m}/\mathbb{Z}_{2m};D_n/D_n)$ | (89) | $4mn$ |
| $DV_3(m,n)$ | $(\mathbb{Z}_{4m}/\mathbb{Z}_{2m};D_n/\mathbb{Z}_{2n})$ | (91) | $4mn$ |
| $DV_3'(m,n)$  $(m,n$ odd$)$ | $(\mathbb{Z}_{4m}/\mathbb{Z}_{m};D_n/\mathbb{Z}_{n})$ | (94) | $2mn$ |
| $DV_4(m,n)$ | $(\mathbb{Z}_{4m}/\mathbb{Z}_{2m};D_{2n}/D_n)$ | (92) | $8mn$ |
| $DV_5(m)$ | $(\mathbb{Z}_{2m}/\mathbb{Z}_{2m};T/T)$ | (90) | $24m$ |
| $DV_8(m)$ | $(\mathbb{Z}_{4m}/\mathbb{Z}_{2m};O/T)$ | (93) | $48m$ |

finite subgroups of U(2) can be found in Theorem 2.2 of [61].

Below we give a construction of the groups that appear in our lists as explicit matrix groups. We begin with the abelian subgroups of U(2). Up to conjugation, they are exactly the groups of the form

$$DV_1(m,n,r,s) = \left\{ e^{\frac{2\pi i x}{mr}} \begin{pmatrix} e^{\frac{2\pi i y}{nr}} & 0 \\ 0 & e^{-\frac{2\pi i y}{nr}} \end{pmatrix} \mid x \in \mathbb{Z}_{mr},\, y \in \mathbb{Z}_{nr},\, x = sy \bmod r \right\}, \quad (84)$$

where $m,n,r,s$ are four integers satisfying

$$\begin{cases} m,n,r \geq 1 \\ m \text{ and } n \text{ have the same parity} \\ r \text{ is even if } m \text{ and } n \text{ are odd} \\ 0 \leq s \leq r/2 \text{ and the greatest common divisor of } s \text{ and } r \text{ is } 1 \end{cases} \quad (85)$$

The order of the group $DV_1(m,n,r,s)$ is $\frac{1}{2}mnr$.

In order to describe the nonabelian groups, we need to introduce the following classical subgroups of SU(2):

- The dihedral group $D_n$, of order $4n$, which can be described as the group generated by two matrices

$$D_n = \left\langle \begin{pmatrix} 0 & i \\ i & 0 \end{pmatrix}, \begin{pmatrix} e^{\frac{i\pi}{n}} & 0 \\ 0 & e^{-\frac{i\pi}{n}} \end{pmatrix} \right\rangle. \quad (86)$$

- The binary tetrahedral group $T$, of order 24, generated by

$$T = \left\langle \begin{pmatrix} 0 & i \\ i & 0 \end{pmatrix}, \frac{1}{2} \begin{pmatrix} 1+i & 1+i \\ -1+i & 1-i \end{pmatrix} \right\rangle. \quad (87)$$

- The binary octahedral group $O$, of order 48, generated by

$$O = \left\langle \frac{1}{\sqrt{2}} \begin{pmatrix} 1+i & 0 \\ 0 & 1-i \end{pmatrix}, \frac{1}{2} \begin{pmatrix} 1+i & 1+i \\ -1+i & 1-i \end{pmatrix} \right\rangle. \quad (88)$$

Using these, we can then describe the Du Val groups. First, we have simple extensions of $D_n$ and $T$ by $\mathbb{Z}_{2m}$:

$$DV_2(m,n) = \left\{ e^{\pi i x/m} y \mid x \in \mathbb{Z}_{2m},\, y \in D_n \right\}, \quad (89)$$

$$DV_5(m) = \left\{ e^{\pi i x/m} y \mid x \in \mathbb{Z}_{2m},\, y \in T \right\}. \quad (90)$$

Then we have extensions by $\mathbb{Z}_{4m}$ of $D_n$, $D_{2n}$, and $O$ with certain restrictions:

$$\mathrm{DV}_3(m,n) = \left\{ e^{\pi i \frac{2x}{2m}} y \;\middle|\; x \in \mathbb{Z}_{2m}, y \in D_n, y^2 = 1 \right\} \cup \left\{ e^{\pi i \frac{2x+1}{2m}} y \;\middle|\; x \in \mathbb{Z}_{2m}, y \in D_n, y^2 \neq 1 \right\}, \tag{91}$$

$$\mathrm{DV}_4(m,n) = \left\{ e^{\pi i \frac{2x}{2m}} y \;\middle|\; x \in \mathbb{Z}_{2m}, y \in D_n \right\} \cup \left\{ e^{\pi i \frac{2x+1}{2m}} y \;\middle|\; x \in \mathbb{Z}_{2m}, y \in D_{2n} - D_n \right\}, \tag{92}$$

$$\mathrm{DV}_8(m) = \left\{ e^{\pi i \frac{2x}{2m}} y \;\middle|\; x \in \mathbb{Z}_{2m}, y \in T \right\} \cup \left\{ e^{\pi i \frac{2x+1}{2m}} y \;\middle|\; x \in \mathbb{Z}_{2m}, y \in O - T \right\}. \tag{93}$$

And finally (note that this group is absent from the original Du Val list, but can be found in [61]),

$$\mathrm{DV}_3'(m,n) = \bigcup_{k=0,1,2,3} \left\{ e^{\pi i \frac{4x+k}{2m}} \begin{pmatrix} 0 & i \\ i & 0 \end{pmatrix}^k \begin{pmatrix} e^{2\pi i y/n} & 0 \\ 0 & e^{-2\pi i y/n} \end{pmatrix} \;\middle|\; x \in \mathbb{Z}_m, y \in \mathbb{Z}_n \right\}. \tag{94}$$

# B  The $G(m,p,r)$ complex reflection groups

For completeness, we give the definition of the infinite family of complete reflection groups $G(m,p,r)$, where $m$, $p$ and $n$ are three positive integers with $p|m$. Let $A(m,p,r)$ be the set of diagonal $r \times r$ matrices $M$ such that

- each diagonal element of $M$ is an $m$-th root of unity, and

- the determinant of $M$ is an $\frac{m}{p}$-th root of unity.

Let $S(r)$ be the set of $r \times r$ permutation matrices. Then

$$G(m,p,r) = \{MP \mid M \in A(m,p,r) \text{ and } P \in S(r)\} \subset \mathrm{U}(r). \tag{95}$$

This group has $\frac{m^r}{p} \times r!$ elements. We are interested in rank $r = 2$, in which case the complex reflection group $G(m,p,2)$ has invariants of degrees $m$ and $\frac{2m}{p}$.

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
