# Peer review of "Classification of all $\mathcal{N}\geq 3$ moduli space orbifold geometries at rank 2"

_SciPost Physics, doi:SciPost Phys. 9, 083 (2020)_

## Round 1 · Referee Report · Noppadol Mekareeya (Referee 1) · 2020-2-18

Strengths

  1. The content of this article is important for a further development in the field of supersymmetric field theories, especially a deeper understanding of $\mathcal{N}=3$ theories in four dimensions.
  2. The analyses and computations were performed with great care.
  3. Despite the technicality of the subject, the material is very well-presented.

Weaknesses

Some minor improvements could be done, such as more explanations on certain results.

Report

The constraints on the moduli space of rank-two $\mathcal{N}=3$ theories were analyzed with great care in this article. The authors studied and classified orbifolds with a triple special Kaehler (TSK) structure that are compatible with $\mathcal{N}=3$ supersymmetry. Using the Hilbert series, the authors then excluded the geometries whose coordinate rings are neither freely generated nor complete intersections. For the geometries that are not excluded, the authors carefully pointed out whether they come from some discrete gaugings, S-folds, or possibly novel theories that have never been studied before. One of the very interesting proposals of this article is probably regarding the moduli space of the form $\mathbb{C}^6/\Gamma'$, which is not a discrete quotient of $\mathbb{C}^6/\Gamma$ for a complex reflection group $\Gamma$. The existence of the moduli space of the form $\mathbb{C}^6/\Gamma'$ could potentially leads to new theories that are not associated to any complex reflection group. This could be very crucial for a deeper understanding of 4d $\mathcal{N}=3$ SCFTs and 3d $\mathcal{N}=6$ SCFTs (upon compactifying the 4d theory on a circle).

Requested changes

  1. In the caption of Table 4 or around that place, the authors should remind the readers again what is the physical reason to exclude the geometries whose coordinate rings are neither freely generated nor complete intersections. (The reasons given in subsection 4.1 were already clear.)
  2. The authors denoted by $(u_1, u_2)$ the fugacities for $SU(3)$, whose character of the fundamental representation can be written as $u_1+u_2 u_1^{-1} + u_2^{-1}$. Observe that each term has to multiply to $1$, as it should be for $SU(3)$. However, in Eq. (4.1), the authors wrote in each factor $u_1$, $u_2 u_1^{-1}$ and $u_2$, which do not multiply to $1$. It would be nice for the authors to justify this in the manuscript.
  3. It would be a good idea to show explicitly the plethystic logarithm $\mathrm{PLog_{\mathcal{M}_\Gamma}}$ for some non-trivial examples somewhere below (4.11) and before subsection 4.1, and make an explicit connection to the coefficient $c_1, \ldots, c_4$ on Page 33. This would make the explanation there much clearer. Moreover, the authors could emphasize whether or not there is any connection between $\mathrm{PLog_{\mathcal{M}_\Gamma}}$ and $\mathrm{PLog_{\Gamma}}$. These two notations are very close to each other. They could potentially lead to a confusion if there is no further explanations.
  4. The Hilbert series presented in Table 4 do not have palindromic numerators. This could imply some information about the coordinate rings of such excluded geometries. For example, by the Stanley theorem, the coordinate rings of such geometries are not Gorenstein (see Section 3.9.5 of Ref. [25]), and the geometries are not Calabi-Yau. Could this be a reason to support the exclusion of such geometries? It would be nice to make some comments about this in the manuscript if possible.

---

## Round 1 · Referee Report · Anonymous (Referee 2) · 2020-2-19

Strengths

1-The approach proposed by the authors does not assume a priori that the Coulomb Branch is freely generated. The validity of this assumption modulo discrete gaugings is controversial and this work provides a new perspective on this point.

Weaknesses

1-The authors need to assume that the moduli space of the theory is an orbifold. At present it is unclear how restrictive this assumption is.

Report

In this paper the authors study and classify orbifold geometries which can be interpreted as moduli spaces of rank two superconformal theories with twelve or more supercharges in four dimensions. The analysis builds on the properties of triple special kahler geometries recently introduced by the same authors and on mathematical results by E. Gottschling published in early sixties.
The resulting list of geometries is further constrained by studying the Hilbert series, which in particular allows the authors to determine whether the candidate theory is a product of rank one theories or includes a free sector. Known properties of rank one theories, which are well understood, can then be effectively exploited to rule out several unphysical cases.
Interestingly, this analysis does not rely on the assumption that the Coulomb Branch of the theory is freely generated, contrary to most approaches currently available in the literature. At present it is believed (although not proven) that non freely-generated Coulomb Branches arise via discrete gaugings only. In this paper the authors find three candidate counterexamples since the corresponding geometries pass all the consistency conditions considered in the paper. This clearly motivates further studies about the above-mentioned conjecture.
Overall the paper is well written and contains interesting results which constitute a good starting point for future investigations on superconformal theories with extended supersymmetry in four dimensions. There are however a few issues the authors should address prior to publication.

Requested changes

1-I believe the authors should add some clarifications regarding the computation of a and c central charges and about the data appearing in table 5. S-folds corresponding to entries 2, 3 and 4 in Table 5 are known to come in two variants, depending on the amount of torsional flux: one is a discrete gauging of U(1) $\mathcal{N}=4$ and the other satisfies the Shapere-Tachikawa formula used by the authors to compute central charges. On the other hand, the only $II^*$ geometry produced by an S-fold construction is a $\mathbb{Z}_6$ gauging of U(1) $\mathcal{N}=4$ and its central charges are those of the parent $\mathcal{N}=4$ theory. This is the reason why the last entry in Table 1 is correctly marked as currently unknown and does not correspond to two D3 branes probing an S-fold. From the paper it seems the authors are claiming that there is a rank one theory with CB operator of dimensions 6 and $a=c=\frac{11}{4}$ coming from an S-fold construction, which is not true. If the authors claim such a theory exists, I think they should provide the corresponding reference. If instead when they discuss $II^*$ geometries they refer to a discretely gauged theory, they should modify accordingly the value of the central charges reported in Table 1.
2-There are a few typos in Table 6. For example entry #10 should be $I_0^*\times III^*$ rather than $I_0^*\times IV^*$.
3-To be consistent with equations (3.1)-(3.3) the symplectic condition after (3.2) should read $MJM^T=J$ and not $M^TJM=J$.

---

## Round 2 · Referee Report · Anonymous (Referee 3) · 2020-11-4

Report

The authors have addressed the comments of the referees. In my opinion, this article deserves publication in SciPost.

---

## Round 2 · Referee Report · Anonymous (Referee 2) · 2020-11-16

Report

The authors have addressed all the issues I have reported. I therefore recommend the paper for publication in SciPost.

---

## Round 2 · Author Response

Dear Editor,

Please find attached the new version of our paper. We have addressed the various comments made by the referees, that we thank for their detailed read and insightful remarks. We have addressed all these comments as detailed below.

---

## Round 2 · List of Changes

1) We have changed the value of the central charge for the Z6 group in Table 1 and updated the caption of that Table accordingly.
2) Below equation (3.2) we have exchanged M and M^T
3) We added a sentence in the caption of Table 4.
3) Below equation (4.3) we have added a sentence.
4) On page 33 we added a full example of computation.
5) In section 5.1 and in Table 5 we have clarified the issue of S-folds, emphasizing that we focus on the full flux theories.

---

## Editorial Decision

published